# Decision-making components and times revealed by the single-trial electroencephalogram

Gabriel Weindel[1,2]*[§], Jelmer P Borst[1][†], Leendert van Maanen[2][†]

[1]University of Groningen, Groningen, Netherlands; [2]Utrecht University, Utrecht, Netherlands

## eLife Assessment

Weindel et al. examine behavioral and EEG data in an innovative contrast comparison paradigm where they vary mean contrast widely while keeping contrast difference constant. As intended, this allowed an elegant decomposition of processing stages: while sensory encoding shortened with increasing contrast in keeping with Pieron's law, the period of decision formation lengthened, in keeping with Fechner's law, which was applied to drift rates in a diffusion model of that period. This is an **important** demonstration of how these two laws apply in concert, to two distinct processing levels, and the multivariate topography parsing, mixed effect models, and diffusion models are **convincing**.

**\*For correspondence:**
gabriel.weindel@gmail.com

[†]These authors contributed equally to this work

**Present address:** [§]Institut de Psychologie, Université de Lausanne, Lausanne, Switzerland

**Competing interest:** The authors declare that no competing interests exist.

**Abstract** Decision-making stems from a sequence of information processing steps between the onset of the stimulus and the response. Despite extensive research, uncertainty remains about the actual cognitive sequence involved that leads to the reaction time. Using the hidden multivariate pattern method, we modeled the single-trial electroencephalogram of participants performing a decision task as a sequence of an unknown number of events estimated as trial-recurrent, time-varying, stable topographies. We provide evidence for five events occurring during participants' decision-making with two visual encoding events and three events capturing respectively attention orientation, decision, and motor execution. This interpretation is supported by the observation that a targeted manipulation of stimulus intensity yields Piéron's law in the interval between encoding and attention orientation, and Fechner's law in the interval between attention orientation and decision commitment. The final decision-related event is represented in the brain as a ramping signal in parietal areas whose timing, amplitude, and build-up predict the participants' decision accuracy.

## Introduction

Making a decision involves several information processing steps within the time from the presentation of a stimulus to the response. The total time required for the completion of each of these processing steps is the reaction time (RT). Specific processes differ between experimental paradigms, but a minimal set that seems to be agreed upon involves encoding of the choice-relevant features of the stimuli, followed by weighting the evidence for each choice, and initiating a response (***Donders, 1868***; ***Ratcliff and McKoon, 2008***; ***Zylberberg et al., 2011***; ***Luce, 1986***). Despite being an almost two century-old problem (***Helmholtz, 1850***), it is unclear how the RT emerges from these putative components.

The answer to this problem has first been hampered by the relatively poor information gained from RT and response-accuracy data alone. Co-registering physiological signals can clarify and extend

conclusions about information processing steps in the RT (*Turner et al., 2017*). Evidence for the putative components that make up RT has been found by registering the electroencephalogram (EEG) during decision tasks. First, a negative deflection in occipital electrodes happening around 200ms after the presentation of a choice, the N200, has been associated with visual encoding of the choice elements by participants (*Nunez et al., 2019*; *Ritter et al., 1979*). Second, EEG data has shown that the weighting of evidence toward the alternatives is associated with a positive voltage developing over centro-parietal electrodes after early visual potentials (*Kutas et al., 1977*). Computational models of decision-making explain the experimental effects observed on these centro-parietal components as an evidence accumulation mechanism (*O'Connell et al., 2012*; *Kelly et al., 2021*). Lastly, a component preceding the response has been shown to lateralize with the side of the executed response (*Coles et al., 1985*). This lateralized readiness potential (LRP) has later been described as arising from an accumulation-to-bound mechanism describing the decision to produce a movement (*Schurger et al., 2012*).

However, the knowledge gained on the nature and latencies of cognitive processes within the stimulus-response interval from such electrophysiological components is limited by the low signal-to-noise ratio of classical neural measurements. To improve the SNR, researchers usually rely on information derived from the averaging of these signals over many trials. Unfortunately, averaging time-varying signals will result in an average waveform that misrepresents the underlying single-trial events (*Luck, 2005*; *Borst and Anderson, 2024*). In the case of decision-making, several studies have shown wide trial-by-trial variation of the timing of cognitively relevant neural events (*Vidaurre et al., 2019*; *Smyrnis et al., 2012*; *Weindel et al., 2021*; *Weindel, 2021*). Furthermore, averaged components are further distorted by the fact that multiple cognitive processes and associated EEG components are typically present within trials and overlap in time between trials (*Woldorff, 1993*), forcing researchers to study physiological components in isolation. A few studies have been able to simultaneously investigate multiple EEG components in decision-making using single-trial approaches. As an example, *Philiastides et al., 2006* used a classifier on the EEG activity of several conditions to show that the strength of an early EEG component was proportional to the strength of the stimulus, while a later component was related to decision difficulty and behavioral performance (see also *Salvador et al., 2022*; *Philiastides and Sajda, 2006*). Furthermore, the authors interpreted that a third EEG component was indicative of the resource allocated to the upcoming decision given the perceived decision difficulty. In their study, they showed that it is possible to use single-trial information to separate cognitive processes within decision-making. Nevertheless, their method requires separate classifiers for each component of interest, limiting the analysis to existing theory of distinct components.

One potential solution mixing both behavior and multivariate analysis of single-trial neural signal to achieve single-trial resolution has emerged through the development of the hidden multivariate methods (*Weindel et al., 2024*; *Anderson et al., 2016*). These methods model the neural data of each trial as a sequence of short-lived multivariate cortex-wide events, repeated at each trial, whose timing varies on a trial-by-trial basis and define the RT. In the case of EEG, it is assumed that any cognitive step involved in the RT is represented by a specific topography recurring across trials. The time jitter in the topography is accounted for by estimating, for each of these events, a trial-wise distribution where the expected time of the peak of the topography is given by the time distribution of the previous event's peak and the expected time distribution of the current event. By constraining, through the recorded behavior, the search for trial-shared sequential activations in the EEG during estimated ranges of time, the hidden multivariate pattern (HMP) model (*Weindel et al., 2024*) provides an estimation of the number of events and their single-trial latency during each trial. Previous similar approaches have shown that different information processing steps can be extracted from the EEG in a wide range of tasks (*Berberyan et al., 2021*; *Zhang et al., 2018*; *Anderson et al., 2016*; *Anderson et al., 2018*; *Krause et al., 2024*). Building on previous work (*van Maanen et al., 2021*), we expect that the EEG data of a decision-making task will be decomposed into task-relevant intervals indexing the information processing steps in the RT. In the current study, we combine this single-trial modeling strategy with strong theoretical expectations regarding the impact of experimental manipulations on the latent information processing steps during decision-making.

The task of the participants was to answer which of two Gabor patches flanking a fixation cross displayed the highest contrast (*Figure 1*, top panel). On a trial-by-trial basis, we manipulated the average contrast of both patches but kept the difference between them constant (see the two

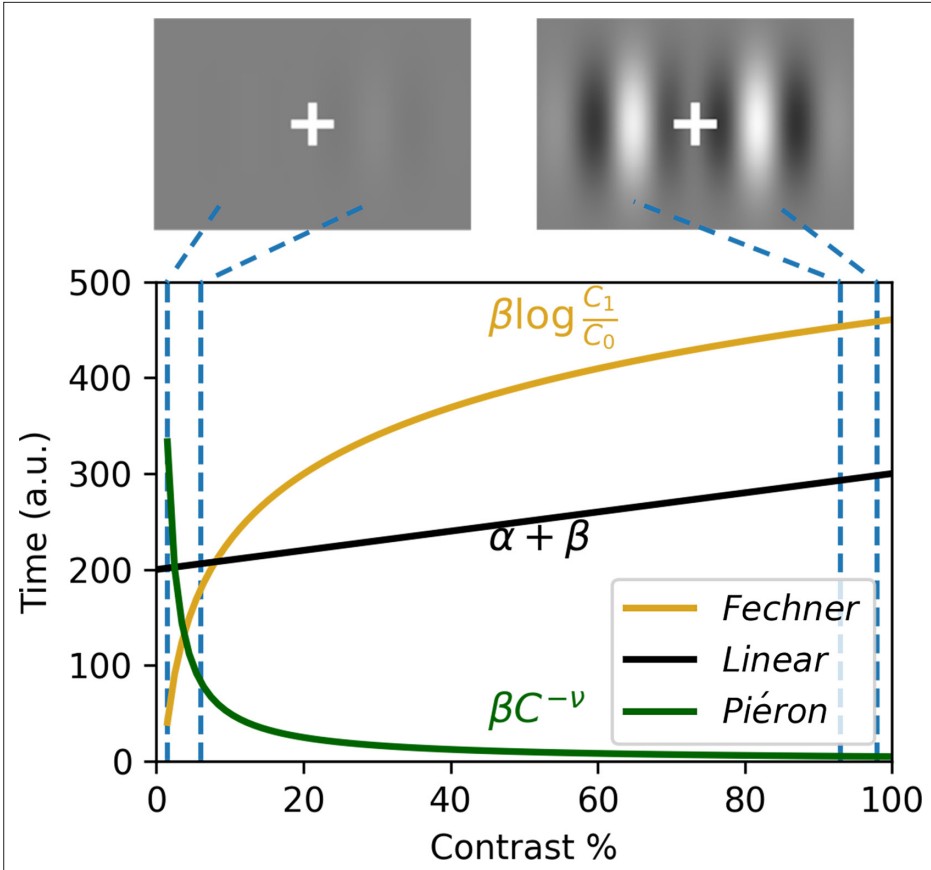

**Figure 1.** Contrast manipulation used in the experiment. Top shows two example stimuli illustrating minimum (left) and maximum (right) contrast values. The bottom panel shows the prediction for the Piéron, the Fechner, and the linear laws for all contrast levels ($C$) used in the study for a fixed set of parameters. The y-axis refers to the time predicted by each law given a contrast value (x-axis) and the chosen set of parameters. $\alpha$, $\beta$, and $\nu$ are respectively the estimated participant-specific intercept, slope, and exponent for the three laws. The Fechner diffusion model additionally includes nondecision and decision threshold parameters (see 'Materials and methods').

example trials in *Figure 1*, one with an average contrast of 5%, and one with an average contrast of 95%, both with a difference of 5%). We hypothesize that this contrast manipulation generates two opposing predictions on encoding and decision processes (*Weindel et al., 2022*) associated with two of the oldest laws in psychophysics: Piéron's law (*Piéron, 1913*) and Fechner's law (*Fechner, 1860*).

Piéron's law predicts that the time to perceive the two stimuli (and thus the choice situation) should follow a negative power law with the stimulus intensity (*Figure 1*, green curve). In contradistinction, Fechner's law states that the perceived difference between the two patches follows the logarithm of the absolute contrast of the two patches (*Figure 1*, yellow curve). As the task of our participants is to judge the contrast difference, Piéron's law should predict the time at which the comparison starts (i.e., the stimuli become perceptible), while Fechner's law should implement the comparison, and thus decision, difficulty. Given that Fechner's law is expected to capture decision difficulty, we connected this law to evidence accumulation models by replacing the rate of accumulation with Fechner's law in the proportional rate diffusion model of *Palmer et al., 2005*. This linking with an evidence accumulation model further allows connecting the RT to the proportion of correct responses. To test the generalizability of our findings and allow comparison to standard decision-making tasks, we also included a speed–accuracy manipulation by asking participants to either focus on the speed or the accuracy of their responses in different experimental blocks.

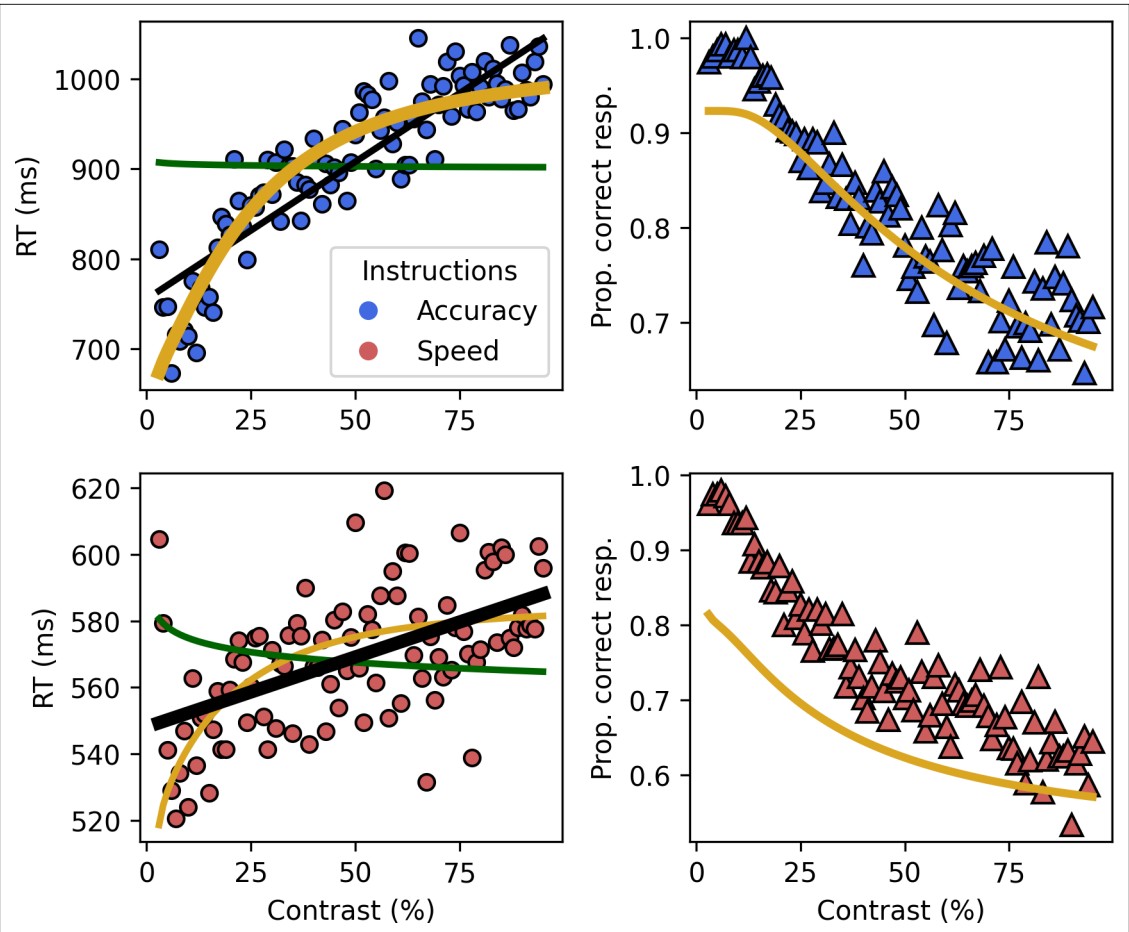

**Figure 2.** Behavioral results partially support Fechner's law. Left: mean RT (dot) for each contrast level and averaged predictions of the individual fits (line) with accuracy (top) and speed (bottom) instructions. Right: mean proportion of correct responses averaged over trials and participant for each contrast level (triangles) along the average predicted proportions for the Fechner diffusion models (line) in accuracy (top) and speed (bottom).

## Results

All results were obtained from the data of 26 participants, with the recorded EEG band-passed filtered between 0.01 and 40 Hz. Trials with an RT faster than 100 ms or slower than 3000 ms or in which an electrode exceeded a rejection threshold of 100 µV during the stimulus-response interval were rejected (3.1% of trials were rejected, see 'Materials and methods' section for a full report). The HMP model assumed that a task-related multivariate pattern event is represented by a half-sine whose timing varies from trial to trial based on a gamma distribution with a shape parameter of 2 and a to-be-estimated scale parameter, controlling the average latency of the event (*Weindel et al., 2024*). Contrary to previous applications (*Anderson et al., 2016*; *Berberyan et al., 2021*; *Zhang et al., 2018*; *Krause et al., 2024*), we assumed that the multivariate pattern was represented by a 25 ms half-sine as our previous research showed that a shorter expected pattern width increases the likelihood of detecting cognitive events (see Appendix B of *Weindel et al., 2024*).

### Perceptual and decision effects compete at the behavioral level

*Figure 2* shows the mean RT and proportion of correct responses. To test what model best described the RT data, we fitted the two opposing psychological laws and an atheoretical linear model. Using a leave-one-out cross-validation strategy, we observed that the best-fitting model is the Fechner diffusion model in the accuracy condition and the linear model in the speed-focused condition (*Table 1*). Having fitted the Fechner law in a diffusion model framework on the RT, we can use the estimated parameters to predict the proportion of correct responses. Despite only estimating parameters on the RT, the prediction of the model is relatively close to the observed proportion of correct responses

**Table 1.** Square root of the mean prediction error (milliseconds) from the leave-one-out procedure for the models applied to the intervals between each event (Ev.) including stimulus onset (S.) and response (R.).
Bold indicates the best model per interval. The $R^2$ in parentheses refers to the fit of the predicted vs. observed mean durations.
Prop. corr. refers to the $R^2$ for the proportion of correct responses observed vs. the proportion predicted by the fit of the Fechner diffusion model to each interval.

| | RT | S.-Ev.1 | Ev.1 - Ev.2 | Ev.2 - Ev.3 | Ev.3 - Ev.4 | Ev.4 - Ev.5 | Ev.5 - R. | Encoding | Decision |
|---|---|---|---|---|---|---|---|---|---|
| **Accuracy** | | | | | | | | | |
| Linear | 44.37 (0.81) | **1.64 (0.13)** | 5.08 (0.17) | 5.78 (0.13) | 42.91 (0.81) | 7.27 (0.37) | **0.68 (0.08)** | 7.26 (0.15) | 45.52 (0.81) |
| Píeron | 89.93 (0.07) | 1.77 (−0.01) | **4.81 (0.27)** | **5.63 (0.17)** | 86.91 (0.07) | 8.85 (0.05) | 0.70 (0.01) | **6.60 (0.30)** | 92.15 (0.08) |
| Fechner | **41.24 (0.85)** | 1.80 (0.00) | 5.55 (0.01) | 6.38 (−0.07) | **39.20 (0.85)** | **7.12 (0.40)** | 0.73 (0.00) | 8.05 (−0.07) | **40.10 (0.86)** |
| Prop. corr. | (0.84) | (−5.75) | (−7.90) | (−6.68) | (0.83) | (−1.76) | (−6.63) | (−7.76) | (0.83) |
| **Speed** | | | | | | | | | |
| Linear | **18.02 (0.39)** | 1.81 (0.14) | 4.19 (0.33) | 5.51 (0.22) | **16.37 (0.49)** | 5.51 (0.33) | **0.66 (0.11)** | 8.18 (0.34) | 18.23 (0.53) |
| Píeron | 23.22 (0.00) | **1.79 (0.14)** | **3.76 (0.47)** | **4.83 (0.38)** | 21.99 (0.07) | 6.44 (0.09) | 0.69 (0.04) | **6.64 (0.57)** | 25.37 (0.07) |
| Fechner | 19.91 (0.27) | 1.90 (0.04) | 4.90 (0.06) | 6.06 (0.05) | 16.89 (0.45) | **5.12 (0.43)** | 0.66 (0.12) | 9.27 (0.12) | **18.16 (0.52)** |
| Prop. corr. | (0.03) | (−3.58) | (−3.61) | (−4.12) | (0.17) | (0.19) | (−2.49) | (−4.73) | (0.66) |

($R^2$=0.84) in the accuracy condition, but the predictive power is very low when speed is emphasized ($R^2$=0.03).

The behavioral results thus tell two stories: first, as long as accuracy was stressed, Fechner's law predicted the RTs and predicted the proportion of correct responses closely. However, when speed was stressed – and consequently one or more of the underlying cognitive processes was shorter – Fechner's law did no longer explain the behavioral data; instead, a linear model fitted the data best. It is unlikely that neither Piéron nor Fechner law impacts the RT in the speed condition. Instead, this result is likely due to the composite nature of the RT where both laws co-exist in the RT but cancel each other out. We explored this in the next sections by applying HMP to the EEG data.

## Five trial-recurrent sequential events occur in the EEG during decisions

Modeling the EEG between stimulus and response as a sequence of multivariate half-sines with a 25 ms duration with varying trial-by-trial latencies revealed that five events are necessary to account for the EEG data. *Figure 3* shows the cumulative probability of occurrences of the events, thereby indicating how variable the latency of the events was over trials. Overlaid on the probabilities, we represent each event at their averaged time location as well as the averaged electrode activity. These topographical maps show that the first event after stimulus onset peaked at around 40 ms with most of its probability of occurrence mass happening before 100 ms (lightest gray curve in *Figure 3*). This component is mainly characterized by a negative deflection on parietal/occipital electrodes and, together with the stimuli having a foveal locus, likely reflects the N40, the earliest visual potentials indicating low-level visual processing (*Proverbio et al., 2021*; *Carretié et al., 2024*). The second event peaked at around 110 ms and presents a strong positivity over occipital electrodes. Given its topography and time, this event probably represents the P100, the first visual potential associated with attention (*Gonzalez et al., 1994*). The third event presented a strong negative polarity on occipital electrodes with a topography and average time of 200 ms. This component can thus be characterized as the N200 (*Ritter et al., 1979*), a potential associated with the onset of evidence accumulation (*Nunez et al., 2019*; *Loughnane et al., 2016*). The fourth and last events share some activity in both parietal and frontal electrodes. Computing the asymmetry of the fourth event (*Figure 3*, bottom) shows a strong negativity over the central electrodes and thus the motor cortex congruent with a right-hand LRP

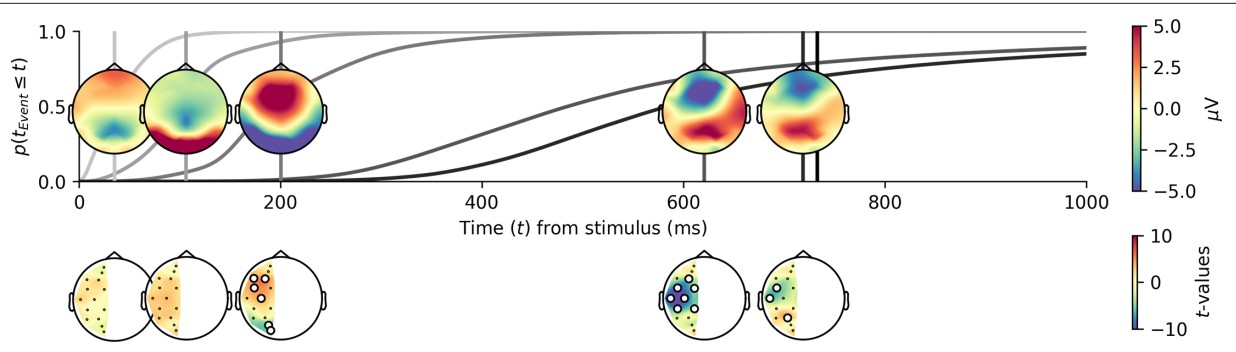

**Figure 3.** Hidden multivariate pattern (HMP) estimation reveals five sequential cognitive events. Top: trial and participant-averaged cumulative event occurrence probabilities (gray lines) for the five detected hidden multivariate events. Overlaid is the average representation of the time location (vertical lines) and electrode contribution of the five identified events based on their trial-by-trial maximum probability. Bottom: t-values from one-sample t-tests on the by-participant differences for each symmetric electrode pair for each event. White dots indicate electrode pairs that deviate significantly from 0 after a Bonferroni correction with a base alpha level of 0.05 (corrected p-value = 0.00071), black dots denote nonsignificant electrode pairs.

(*Coles et al., 1985*). The timing of the last event, a few milliseconds before the response, as well as its strong parietal positivity, suggests that this event is the centro-parietal positivity (CPP) reported to track the accumulation of evidence (*O'Connell et al., 2012*; *Kelly et al., 2021*).

## Inter-event times align with encoding and decision laws

If the detected events capture processes relevant to the decision-making function, we believe that their time of occurrence should be differentially affected by the contrast manipulation. To support this, we fitted Piéron's and Fechner's law as well as a linear model to the by-participant-averaged inter-event durations, as a function of contrast (*Figure 4* and *Table 1*) and speed–accuracy instructions. As for the behavioral data, the best-fitting model was evaluated based on a leave-one-out procedure. To conduct statistical inferences on the effect of the speed–accuracy instructions and contrast, we also fitted Bayesian linear mixed models on the trial-wise event duration in milliseconds (*Table 2*, see 'Materials and methods').

Regarding the effect of contrast on duration, the first interval from stimulus to the first event was best explained by a linear model in Accuracy but Piéron's law in the speed condition. The linear mixed model on this interval suggests a weak effect of contrast in the speed instruction (contrast coefficient, *Table 2*), but the credible intervals included 0. The intervals from Event 1 to Event 2 as well as Event 2 to Event 3 were both best explained by Piéron's law in both speed–accuracy instructions and displayed a negative effect of contrast in the linear mixed models. The interval from Event 3 to Event 4 was best explained by Fechner's law in accuracy but not in speed. The linear mixed coefficient does, however, suggest that an increase in the duration with an increase of contrast is also present in the speed condition. The interval between the detected HMP Event 4 and Event 5 is best explained by Fechner's law in both speed and accuracy instructions with a clear positive effect of contrast on the latency of this interval. Regarding the effect of speed–accuracy conditions, as shown by the linear mixed models (SAT coefficient, *Table 2*), all intervals are affected by the instructions to the notable exception of the interval from the stimulus onset to the first event. The observation that most intervals are impacted by speed–accuracy instructions is congruent with previous reports using alternative methodologies and multimodal recording methods or intracranial measurements (*Steinemann et al., 2018*; *Weindel et al., 2021*; *Heitz and Schall, 2012*).

Because several intervals conformed to Pièron's and Fechner's law, we combined the intervals that displayed a negative relationship with contrast (intervals from Event 1 to Event 2 and Event 2 to 3, labeled as 'encoding') and those that displayed a positive relationship with contrast (intervals from Event 3 to Event 4 and Event 4 to 5, labeled as 'decision') based on the linear mixed model coefficients (*Table 2*). The encoding interval is best explained by Piéron's law ($R^2$ Piéron: speed = 0.53, accuracy = 0.34, *Figure 4B*, *Table 1*). The observation that an early interval that seems associated with perception fits Piéron's law is congruent with earlier empirical observations that Piéron's law is associated with stimulus detection (*Piéron, 1913*; *Banks, 1973*; *Chocholle, 1940*; *Overbosch et al., 1989*;

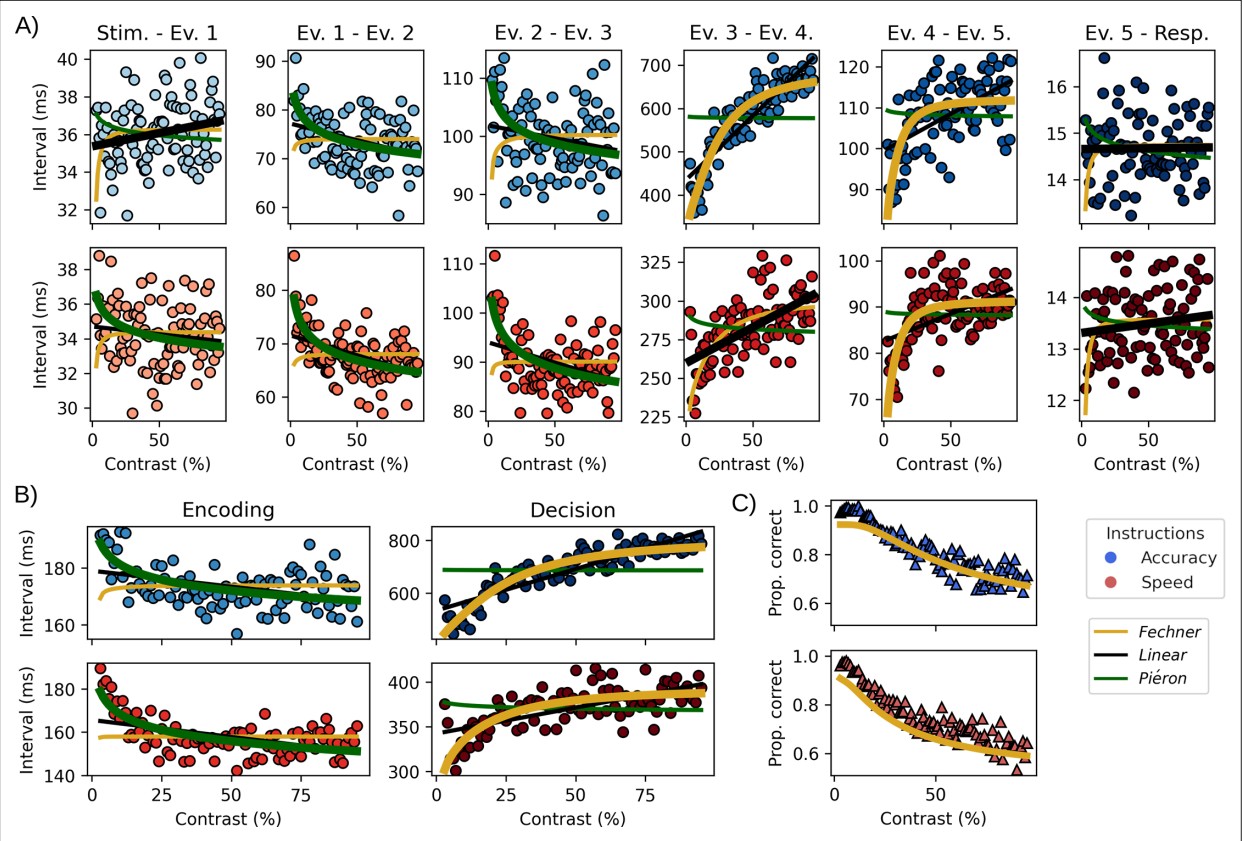

**Figure 4.** Inter-event interval as a function of experimental factors. (**A**) The panels represent the average duration between events for each contrast level, averaged across participants and trials (stimulus and response respectively as first and last events) for accuracy (top) and speed instructions (bottom). The lines represent the fits for the linear (black), Piéron (green), and Fechner (yellow) models. Winning model lines are represented as thicker than the other for each panel. (**B**) Same as in (**A**) but with similar intervals summed together to form encoding (Ev.1-Ev2+Ev.2-Ev.3) and decision (Ev.3-Ev4+Ev.4-Ev.5) times. (**C**) Observed proportion of correct responses (triangles) compared to the prediction (lines) from the Fechner diffusion model fit to the decision interval in (**B**).

*Bonnet et al., 1999*). Note that this observation is not necessarily incongruent with theoretical work that argued that Piéron's law could also be a result of a response selection mechanism (*Stafford and Gurney, 2004*; *van Maanen et al., 2012*; *Palmer et al., 2005*). It could be that differences in stimulus intensity between the two options also contribute to a Piéron-like relationship in the later intervals,

**Table 2.** Summary of the linear mixed models coefficients with the mean and the 95% credible intervals from the posterior distributions for the linear mixed models on intervals for each event (Ev.) including stimulus onset (S.) and response (R.).

| Interval | Intercept | Contrast | SAT | Contrast × SAT |
|---|---|---|---|---|
| S.-Ev.1 | 34.69 (33.25, 36.15) | -0.90 (−2.49, 0.60) | 0.61 (−0.64, 1.73) | 2.36 (0.41, 4.55) |
| Ev.1–Ev.2 | 71.66 (67.57, 75.62) | −7.49 (−11.38, −3.86) | 5.65 (3.32, 8.06) | 0.67 (−3.98, 5.39) |
| Ev.2–Ev.3 | 93.97 (88.72, 99.61) | −8.10 (−12.87, -2.64) | 7.80 (3.80, 11.82) | 3.68 (−2.57, 10.35) |
| Ev.3–Ev.4 | 260.10 (225.28, 292.15) | 46.07 (9.27, 86.43) | 175.28 (121.63, 233.48) | 249.76 (154.58, 344.71) |
| Ev.4–Ev.5 | 82.48 (76.53, 88.85) | 11.88 (6.16, 17.51) | 16.85 (11.93, 22.27) | 6.21 (−1.35, 13.13) |
| Ev.5–R. | 13.31 (12.77, 13.84) | 0.36 (-0.19, 0.89) | 1.34 (0.91, 1.79) | −0.32 (−1.12, 0.52) |

that is convoluted with Fechner's law (see *Donkin and Van Maanen, 2014*, for a similar argument). Unfortunately, our data do not allow us to discriminate between a pure logarithmic growth function and one that is mediated by a decreasing power function.

The decision interval is best explained by Fechner's law ($R^2$ Fechner: speed = 0.52, accuracy = 0.86). Using the drift rate and decision boundaries estimated from the fit of a Fechner diffusion model (see 'Materials and methods') on the decision durations, we can predict the proportion of correct responses for each interval. This shows that the decision interval predicts the participants' proportion of correct responses (*Figure 4C*) in accuracy ($R^2 = 0.83$) but also in speed ($R^2 = 0.66$), contrary to the behavioral fits (*Figure 2*).

## Functional interpretation of the events

So far, we have shown the congruence between the sequential events detected in the stimulus-response interval and known event-related potentials (ERPs): N40, P100, N200, LRP, and CPP. We also showed that the contrast manipulation impacted the times between the identified EEG events (*Figure 4* and *Table 1*) as expected by two century-old psychophysical laws. The dynamics of each event, both in relation to the contrast manipulation as well as their build-up, can be instrumental in interpreting their functional role (*O'Connell et al., 2012*; *Coles et al., 1985*; *Kelly et al., 2021*). One traditional representation of the time course of specific events is to draw the trial-averaged voltage time course for a set of electrodes. *Figure 5A* represents, for three typical sets of electrodes (top panels), the average ERP for each speed–accuracy instruction and the contrast binned in 10 equally spaced bins. Gray lines show the average occurrence of the sequential events. To test for a significant effect of contrast, we performed a nonparametric cluster-level paired *t*-test for the participant-averaged time series for low (< 50%) vs. high (> 50%) contrasts in both speed–accuracy instructions separately. This representation shows that there is an inversion of the contrast effect with higher contrasts having a higher amplitude on the electrodes associated with visual potentials in the first couple of deciseconds (left panel of *Figure 5A*) while parietal and frontal electrodes show a higher amplitude for lower contrasts in later portions of the ERPs (middle and right panels of *Figure 5A*). Nevertheless, identifying neural dynamics on these ERPs centered on stimulus is complicated by the time variation of the underlying single-trial events (see probabilities displayed in *Figure 3* for an illustration and *Burle et al., 2008* for a discussion). The likely impact of contrast on both amplitude and time on the underlying single-trial event does not allow for interpreting the average ERP traces as showing an effect in one or the other dimension without strong assumptions (*Luck, 2005*). If, as estimated by the HMP method, we access the time of single-trial events, we can instead time-lock electrode activity at the time of the event and analyze amplitude by controlling for time variation of the single-trial events.

We thus inspected how the signal recorded with the EEG electrodes varied with contrast levels surrounding the trial-by-trial most likely peak time of each event. Instead of selecting electrodes, we used each event's topography as a spatial filter by computing the dot product between the topographies of each event (i.e., the average of the electrode activity at each event's most likely single-trial time) and the electrode signal at all time samples from stimulus onset to 100 ms after each event (*Figure 5*). This results in single-trial time series of the match between the EEG signal and that event's topography. We then centered each trial and each event's time series to the trial most-likely time of the event and averaged the time series separately for the two speed–accuracy instructions and 10 contrast bins. This procedure shows that, across both speed–accuracy instructions, the first event, interpreted as the N40, is expressed as a narrow band around the single-trial most-likely time of that event as shown by the single-trial surface plot (*Figure 5C*, first column). The trial-averaged traces show that its expression in the EEG is not linked to a contrast difference before 15 ms after its peak and that this effect might be linked to the peak of the following P100 (see black lines in the surface plot of the first column in *Figure 5C*). The second event, the P100, is clearly affected by the contrast manipulation with a significant effect of contrast around and during its peak. The peak of the third event, interpreted as a N200, does not seem to vary with contrast. Instead, a period preceding the N200 peak is significantly different for the contrast levels, but given that the match is negative, it is likely attributable to the preceding P100 as their topographies are mostly anticorrelated. Regarding the fourth event (before the last column of *Figure 5C*), interpreted as the LRP, it can be seen that the event is mostly happening abruptly in the time course. The effect of contrast is not very clearly defined but seems to happen either right at the peak or just after, and, in this case, the effect of contrast is

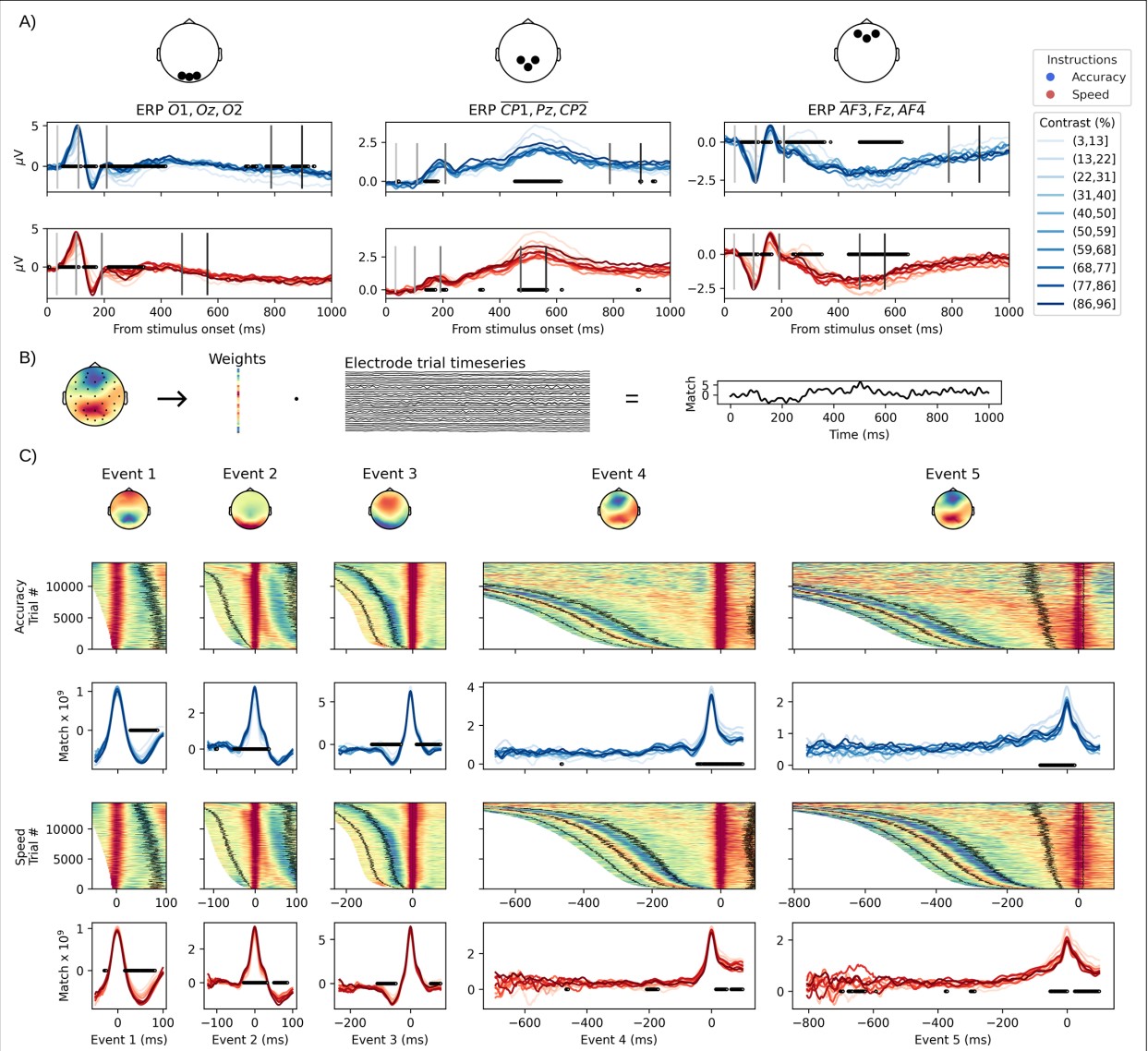

**Figure 5.** Electrophysiological analysis of the time leading to each event. (**A**) Stimulus-centered event-related potentials (ERPs) obtained by averaging three electrodes in occipital, parietal, and frontal areas across trials and participants for 10 binned contrast levels in both speed–accuracy instructions. Vertical lines indicate the average peak time of each hidden multivariate pattern (HMP) event (later = darker). To statistically test for an effect of contrast, we performed a nonparametric cluster-level paired *t*-test for high (> 50%) vs. low (< 50%) contrasts on the participant-averaged waveforms. Significant clusters are marked as black dots at the corresponding times in each panel. (**B**) Illustration of the event match computation. A single-trial match time series (right-most panel) is obtained by taking the dot product between the single-trial electrode time series and the weights of an events' average topography. (**C**) HMP detected events time courses obtained by using the average topographies (top panel) as a spatial filter for the stimulus onset to 100 ms after the single-trial event peak time. For each speed–accuracy instruction, the data is presented at the single-trial level (surface plot), with trials sorted by the single trial most likely peak time of the event, and averaged across trials for 10 contrast bins after aligning to the most likely peak time of the event (waveform plot). On the surface plot, a z-scoring was performed after applying a Gaussian window over 50 trials and with a standard deviation of 20. Black lines represent the peak(s) of previous or following events, relative to the event used as spatial filter, obtained from a rolling mean over 50 trials.

that lower contrast levels yield a higher expression of that particular topography. It is to be noted that some periods of significance of the contrast effect are observed much earlier than the peak of the LRP but without a discernible pattern. The fact that these periods indicate moments in the time course where higher contrasts yield higher amplitude of the component links these periods to the overlap of the preceding detected sensory events. For the last event, the CPP, instead of an abrupt onset as in the preceding LRP, there seems to be a progressive build-up of the match as suggested by

the surface plot and indicated by the significant periods preceding the peak of the event (last column of *Figure 5C*). This effect is in the expected direction for a decision variable (lower contrast being easier) and the duration of this period appears shorter in the speed than in the accuracy condition. This is congruent with an account of a CPP building up as a ramp before the decision commitment

## Box 1. Testing single-trial CPP build-up.

The idea of electrophysiological signals reflecting evidence accumulation has received critical considerations (*Latimer et al., 2015*; *Frömer et al., 2024*; *Zoltowski et al., 2019*). If the CPP event detected by HMP is representing evidence being accumulated, then its trial-by-trial build-up should vary with contrast levels and be related to the proportion of correct responses. As a proxy for this build-up, we first computed the match (as illustrated in *Figure 5B*) between the topography of the CPP and the samples in between the trial estimated peak of that event and the 250 ms period preceding it. We then performed a linear regression for each trial and first applied a linear mixed model (see 'Materials and methods' for model specification) on the intercepts and slopes obtained. On the obtained single-trial intercepts (mean = 1.68, 95% CrI = [1.15, 2.18]), we observed a significant effect of contrast (mean = –0.52, 95% CrI = [–0.85, –0.16]) but not SAT (mean = –0.004, 95% CrI = [–0.007, –0.001]) nor the interaction with contrast (mean = –0.22, 95% CrI = [–0.59, 0.13]). For the single-trial slopes (mean = 0.015, 95% CrI = [0.011, 0.018]), we observed a significant effect of contrast (mean = –0.008, 95% CrI = [–0.010, –0.006]) and SAT (mean = –0.004, 95% CrI = [–0.007, –0.001]) but not their interaction (mean = 0.000, 95% CrI = [–0.003, 0.003]). Secondly, we computed the Spearman correlation coefficient between the contrast-averaged proportion of correct responses and either the intercepts or slopes of the linear regression models, averaged over participants, for each contrast level.

On the single-trial slopes of the CPP event build-up, these observations support the hypothesis that a ramping signal, congruent with evidence accumulation, is developing prior to the response of the participants. Furthermore, assuming that the intercepts of the linear regressions reflect the value of evidence at choice commitment, the observation that the intercept predicts the proportion of correct response supports a decision-making model with a decision threshold dynamically decreasing over time (*Drugowitsch et al., 2012*; *Tajima et al., 2016*; *van Maanen et al., 2016*). This is because under the assumption of a decreasing threshold, a lower rate of evidence accumulation will cross the decision threshold at a lower value, here represented by a smaller intercept.

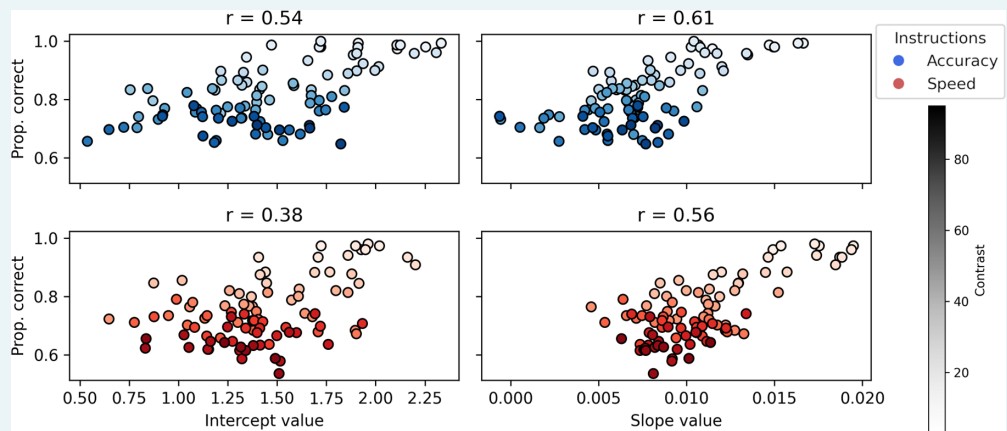

**Box 1—figure 1.** Both the intercepts and slopes of the linear models predict the proportion of correct responses with correlation coefficients ranging from 0.38 to 0.61 (all p-values < 0.001).

with a peak varying with speed stress (*Kelly et al., 2021*). To further test how this electrophysiological build-up relates to behavior, we tested their relation in *Box 1*.

## Discussion

By modeling the recorded single-trial EEG signal between stimulus onset and response as a sequence of multivariate events with varying trial-by-trial peak times, we aimed to detect recurrent events that contribute to the duration of the RT in the present perceptual decision-making task. In addition to the number of events, using this HMP approach (*Weindel et al., 2024*), we estimated the trial-by-trial probability of each event's peak, therefore assessing at which time sample each event was the most likely to occur. To identify specific cognitive processes, we combined this approach with a targeted manipulation of stimulus intensity expected to elicit two well-known psychophysical laws relating to stimulus encoding and decision, respectively Piéron's and Fechner's laws. In this study, we detected five events in the RT of participants performing a simple visual decision-making task. We used each event's most likely time to extract the trial-by-trial interval between successive events to categorize these periods as being part of encoding (Piéron's law) and decision times (Fechner's law) depending on the impact of the stimulus intensity manipulation (*Figure 4*). We then used the average electrode values at these trial-by-trial most likely times as spatial filters to identify the dynamics of each event and their sensitivity to the stimulus intensity manipulation (*Figure 5*). Relating inter-event intervals and electrode activity to the psychophysical laws allowed us to unambiguously map the expected cognitive processing steps to the events discovered in the single-trial EEG between the stimulus onset and the response.

### Visual encoding time

The first event after stimulus onset, identified as the N40, most likely represents the first EEG identifiable visual processing event, probably originating from subcortical structures such as the lateral geniculate nucleus (*Proverbio et al., 2021*). The following event, the P100, is expressed around 70 ms after the N40, its topography is congruent with reports for stimuli with low spatial frequencies as used in the current study (*Kenemans et al., 2002*; *Kenemans et al., 2000*; *Proverbio et al., 1996*). The timing of this P100 component is changed by the contrast of the stimulus in the direction expected by Piéron's law (*Figure 4A*). The expression of this component in the EEG (*Figure 5*) also varies congruently with stimulus intensity, with higher contrasts yielding a higher P100. Given all these observations, it is likely that this P100 represents the visual encoding time of the stimuli. Interestingly, another component with a close topography seems to be expressed around the P100 as suggested by the inverted positive sigmoid-like trend in the surface plot of the second column in *Figure 5C*. Based on its timing and the observation that its topography is close to the P100, it could represent a C1 (*Jeffreys and Axford, 1972*). This component might also be sensitive to contrast as revealed by the significant periods of difference before and after the P100 peak. Nevertheless, this event, seemingly overlapping with the P100 even at the trial level (*Figure 5C*), cannot be recovered by the method we applied. The fact that the P100 was recovered instead of the C1 could indicate that only the timing of the P100 contributes to the RT (see Section 3 of *Weindel et al., 2024*).

### Attention orientation

After the P100 and a duration again influenced by the stimulus contrast according to Piéron's law (*Figure 4*), the third event peaks with a topography that resembles the N200 (*Figure 3*), a component linked to figure-ground separation (*Ritter et al., 1979*). The timing of this event changes with contrast congruently to the expectation of Piéron's law and dictates the onset of decision-making (*Figure 4*) congruent with previous studies (*Nunez et al., 2019*; *Loughnane et al., 2016*). It is interesting to note that, while this N200 peak time respective to the P100 is linked to visual processing, we did not find evidence for visual or decision-related effects in the EEG expression of that event (*Figure 5*). We do, however, observe an asymmetry in the topographical map *Figure 3*. This asymmetry might point to an attentional bias with participants (or at least some participants) allocating attention to one side over the other in the same way as the N2pc component (*Luck and Hillyard, 1994*). Based on this collection of observations, we conclude that this third event represents an attention orientation process. In line with the finding of *Philiastides et al., 2006*, this attention orientation event might also relate to the

allocation of resources. Other designs varying the expected cognitive load or spatial attention could help in further interpreting the functional role of this third event.

## Motor execution

The before-last event, identified as the LRP, shows a strong hemispheric asymmetry congruent with a right hand response. The peak of this event is approximately 100 ms before the response, which is also congruent with reports that the LRP peaks at the onset of electromyographical activity in the effector muscle (*Burle et al., 2004*), typically happening 100 ms before the response in such decision-making tasks (*Weindel et al., 2021*). Furthermore, while its peak time is dependent on contrast, its expression in the EEG is less clearly related to the contrast manipulation than the following CPP event. Adding the abrupt onset of this potential, we believe that this event is the start of motor execution engaged after a certain amount of evidence. The evidence for this interpretation is manifest in the fact that the event's topography shares some activity with the CPP event that follows, an expected result if the LRP is triggered at a certain amount of evidence, indexed by the CPP. This interpretation bears the prediction that the start of motor execution does not end the decision, which is supported by the observation that the duration between the LRP and the following CPP also conforms to Fechner's law. Additional support comes from a growing number of studies showing changes of mind in similar decision-making tasks even after the onset of electromyographical activity in the effector muscle (*Spieser et al., 2017*; *Weindel et al., 2021*; *Gajdos et al., 2019*; *Weindel, 2021*). Based on all these observations, it is therefore very likely that this LRP event signs the first passage of a two-step decision process as suggested by recent decision-making models (*Servant et al., 2021*; *Verdonck et al., 2021*; *Balsdon et al., 2023*).

## Decision event

Finally, the last event, the CPP, terminates the decision as shown by the observation that the time from the N200 to this last event peak is best fitted by a Fechner diffusion model and that the estimated parameters predict the proportion of correct responses at a high accuracy ($R^2 \geq 0.66$). Regarding the build-up of this component, the CPP is seen as originating from single-trial ramping EEG activities, but other work (*Latimer et al., 2015*; *Zoltowski et al., 2019*) has found support for a discrete event at the trial level. The ERPs on the trial-by-trial centered event in *Figure 5* can be explained under both theoretical accounts. As outlined above, the LRP is indeed a short burst-like activity, but the build-up of the CPP between high vs. low contrast diverges much earlier than its peak. Furthermore, *Box 1* shows that both the intercept (value at the events' peak) and the slope (build-up of the signal) of single-trial linear regression models fitted on the match between the CPP and the activity prior to the peak are impacted by the contrast as expected for an evidence accumulation signal and predict participants' accuracies. This means that the rate of build-up of the CPP as well as the value before the response contain information on the likelihood of the participant making a correct response. Finally, simulating EEG data with a ramping function, as expected from evidence accumulation models, shows that the ERP profile for such a simulation (Appendix 1) is again compatible with the detected CPP as a ramp. Thus, as proposed using intracranial recordings in monkeys (*Stine et al., 2023*), we concur that decision-related signals can be best explained by both a progressive accumulation predicting correctness and an abrupt response execution event.

## Generalization and limitations

The present study showed what cognitive processes are contributing to the RT and estimated single-trial times of these processes for this specific perceptual decision-making task. The identified processes and topographies ought to be dependent on the task and even the stimuli (e.g., sensory events will change with the sensory modality). More complex designs might generate a higher number of cognitive processes (e.g., memory retrieval from a cue, *Anderson et al., 2016*) and so could more natural stimuli which might trigger other processes in the EEG (e.g., appraisal vs. choice as shown by *Frömer et al., 2024*). Nevertheless, the observation of early sensory vs. late decision EEG components is likely to generalize across many stimuli and tasks as it has been observed in other designs and methods (*Philiastides et al., 2006*; *Philiastides and Sajda, 2006*; *Salvador et al., 2022*). To these studies we add that we can evaluate the trial-level contribution, as already done for specific processes (e.g., *Si et al., 2020*; *Sturm et al., 2016*) for the collection of events detected in the current study.

While this represents, to the best of our knowledge, an unprecedented precision in the description and single-trial estimation of the timing of cognitive processes, some limitations are inherent to the method used. An HMP method assumes that cognitive events are sequential to one another; therefore, the estimated solution cannot account for all events in the EEG. If two components are close in time and overlapping at the trial level (e.g., C1/P100 discussed in the visual encoding time section), only the event that has both the strongest signal and is most linked to the preceding and following event will be detected. Additionally, the current study fit the same model across all participants and conditions. It cannot be excluded that inter-individual differences in signal-to-noise ratio, events' topography, and sequence composition (i.e., cognitive processes, *Archambeau et al., 2023*; *van Maanen et al., 2014*) or time courses (e.g., partial responses, see *Gajdos et al., 2019*; *Weindel, 2021*) could lead to incorrect estimations. Critically, an HMP model assumes that an event is present across all trials, thereby providing an estimate of an event's position even if this event is absent for a particular trial/participant (see simulations in Section 3 of *Weindel et al., 2024*). Second-order analyses are possible to assess the presence of an event (e.g., to evaluate different strategies; *Den Otter et al., 2025*) and are the topic of future work.

## Conclusion

To the question of the processing steps in the RT raised by *Donders, 1868* and *Helmholtz, 1850* more than a century ago, we can affirm that the RT in this decision-making task is made up of five main steps: early and late visual processing, attention orientation, response execution, and decision-making. This decomposition of the RT into processing steps allowed us to explicitly test the impact of experimental conditions on these different intervals. Additionally, the trial-wise resolution allows for straightforwardly using the extracted times to address what theory best explains observed behavior (*Turner et al., 2017*; *Ghaderi-Kangavari et al., 2022*) or apply advanced signal processing methods while accounting for trial-wise variation in time. This likely represents a critical step for a modern approach to mental chronometry in which the analysis of neural data is constrained by the behavior and in turn informs on the generative model of behavior (*Meyer et al., 1988*).

## Materials and methods

### Participants

26 participants with normal or corrected-to-normal vision and no history of neurological diseases signed an informed consent form to participate in the study. The study was validated by the CETO ethical committee of the University of Groningen (ID 94056673). All participants received a compensation of 12 EUR for a session of 90 minutes including EEG set-up.

### Task

The participants performed the experiment in a shielded windowless EEG lab. They were seated approximately 90 cm from a 24-inch LCD screen with a refresh rate of 60 Hz. Participants indicated their response using the index and middle fingers of their right hand on the 'n' and 'm' keys, respectively, for left and right responses. The inter-trial interval from the response to trial $n$ to the stimulus of trial $n + 1$ was randomly sampled from a uniform distribution between 0.5 and 1.25 seconds. Self-paced breaks were interspersed every 140 trials. Participants were instructed to keep their gaze on the fixation cross during the blocks of trials. Speed and accuracy feedback was provided at the end of each block with eventual oral feedback in case the responses were not fast or accurate depending on the SAT instruction. The SAT instructions were displayed at the beginning of each block and varied every two blocks of trials. Participants were trained on the task and on the SAT instructions for 10 minutes with a short block of alternating SAT instructions with corresponding feedback.

Stimuli were controlled by the *psychopy* Python package (*Peirce, 2007*). The stimuli consisted of two Gabor patches on the left and right of a fixation cross. The gratings had a size of 2.5 degrees of visual angle and a horizontal spatial frequency of 1.2 cycles per degree. The contrast was manipulated by setting a contrast value for both gratings and subtracting 2.5% contrast on the incorrect side and adding 2.5% on the correct side. For each trial, the initial contrast was sampled from a uniform distribution with a minimum of 3.5% contrast to a maximum of 95.5% of contrast. The timing of the stimulus was corrected based on a photodiode.

## EEG

EEG data were acquired from 32 electrodes using a BioSemi Active Two system at a sampling rate of 1024 Hz. Additionally, two electrodes were placed at each outer canthus and one below the left eye to record horizontal and vertical (by combining with Fp1) electro-oculograph. Two electrodes were placed on the mastoid bones for reference during acquisition only and for inspection of bad electrodes. For most of the participants, scalp impedance of the electrodes was kept at $< 20k\Omega$, except for two participants where it was kept at $< 30k\Omega$. The data was recorded using the BioSemi acquisition program ActiView 811.

All preprocessing was done using the *mne* package (v1.9.0) (*Gramfort et al., 2014*) with a semi-automated preprocessing custom script (see the Github repository). First, a template montage for the EEG electrode positions was applied, and the signal was visually inspected for bad electrodes (9 in total for the 26 participants). The trigger indicating the time of stimulus onset was corrected based on the signal of a co-registered photodiode. The signal was then low-pass filtered at 40 Hz (to avoid rejection by μV threshold of high-frequency activity, see next section) and re-referenced to the average of the electrodes. An ICA was performed to remove ocular-related activity on a 256 Hz downsampled version, bandpass filtered between 1 and 80 Hz and epoched based on stimulus onsets. The independent component rejection was partially automated by using the correlation of the IC time course to the vertical and horizontal EOG. Visual inspection of the independent components was used to ensure a proper classification. On average, 1.88 components were removed (max = 3). The rejected ICs were zeroed out on the low-pass only filtered signal and (eventual) bad electrodes were then interpolated using spherical splines. The epochs were then created by using 200 ms before the stimulus as baseline correction and keeping samples up to 3000 ms after stimulus.

## Analyses

Trials in which an electrode exceeded 100 μV between stimulus and response were discarded. Trials with an RT faster than 100 ms, or longer than the epoching window of 3 seconds, were discarded. Overall, 3.1% of the recorded trials were rejected for all the analyses presented in the paper. All analyses were performed using Python (v3.12.3). All RTs were given an offset of 10 ms to avoid edge effects in the cross-correlation involved by HMP.

### Hidden multivariate pattern

An HMP model estimates the position of a pattern (e.g., a half-sine with a fixed duration) in time at each trial based on the combination of a time probability distribution, typically a gamma distribution with a shape of 2 and a scale free-to-vary, and a multivariate spatial distribution (electrode or principal component contribution to the pattern). In the case of several events, an HMP model assumes that they are sequential, thus the estimation of the n−1th event influences the estimation of the *n*th event in addition to that event's time and space parameters. The parameters are estimated iteratively through an expectation maximization algorithm (see *Weindel et al., 2024* for a full description).

The HMP analyses followed the pipeline suggested in *Weindel et al., 2024* and applied using the *hmp* python package (v1.0.0-b.2). The EEG data was first truncated to the data between stimulus onset and response. A PCA was applied to the average of the participant variance–covariance matrices. The 15 first principal components (PCs), explaining more than 99.9% of the variance, were retained. The resulting time series was z-scored per participant by subtracting the participant mean and dividing by the participant standard deviation for each trial of all combined PCs. The resulting dataset was used in the HMP estimation. The HMP model was fit with the expectation of a 25 ms half-sine. The number of events was estimated using the standard cumulative fit routine with the default parameters of the associated Python package. We fitted the same model to the data of all participants across all conditions.

### Asymmetry of the event topographies

To compute the asymmetries presented in *Figure 3*, we first calculated the electrode values for each trial at the most likely event times, averaging these per participant and event. We then subtracted the value of each electrode in the right hemisphere from its counterpart in the left hemisphere (e.g., Fp1 - Fp2) for each participant and event and used a paired one-sample *t*-test to determine whether the

difference between the two electrodes was significant per event. To correct for multiple comparisons, we adjusted the initial p-value of 0.05 for the 70 tests performed (14 electrodes × 5 events).

## Linear mixed models

The linear mixed models on latencies were performed on the raw milliseconds scale and included SAT instructions, contrast, and their interaction as predictors. The maximum random effect structure given the fixed effects formulation was chosen. Predictors were coded with SAT predictor as treatment contrasted with speed as 0 and accuracy as 1, contrast was treated as numerical from 0 to 1. To ensure convergence, the models were fitted using a Bayesian estimation with 4 MCMC chains and 2000 samples (1000 as tuning samples) using the bambi Python package (v0.15.0 *Capretto et al., 2022*) with default priors weakly informed by the data. Posterior distribution was summarized with the point estimate of the maximum a posteriori and 95% credible interval (CrI) computed with the Arviz Python package (v0.21.0; *Kumar et al., 2019*) based on the highest density interval (*Kruschke, 2010*).

## Permutation cluster analysis

To test the significance of the time steps in the ERP analyses, we first computed the time series of each event (as illustrated in *Figure 5*) for all trials from the time of stimulus onset up to 100 ms after the peak of the event. For convenience in the use of the statistical test, the contrast conditions were split to aggregate values lower and higher than 50%. To control for the multiple comparisons involved by the significance test at each time point, we used a nonparametric cluster-level paired *t*-test as implemented in the *mne* package (*Gramfort et al., 2014*) using the default values (version 1.8.0). Rejecting the null hypothesis indicates that the cluster structure in each condition (here low vs. high contrast) is not exchangeable (*Sassenhagen and Draschkow, 2019*).

## Model fits

Piéron's law (*Piéron, 1913*) predicts that the time to perceive ($T_P$) will be negatively related to the contrast ($C$) values (*Figure 1*):

$$T_P = \beta C^{-\alpha} \tag{1}$$

With a modality/individual specific adjustment slope ($\beta$) and exponent ($\alpha$). In contradistinction, Fechner's law (*Fechner, 1860*) states that the perceived difference ($p$) between the two patches follows the logarithm of the ratio of physical intensity between the contrasts of the two patches:

$$p = \beta \log \frac{C_1}{C_0} \tag{2}$$

where $C_1 = C_0 + \delta$, again with a modality and individual specific adjustment slope ($\beta$). By connecting this perceived difference to a decision model with an accumulation to bound mechanism (*Palmer et al., 2005*), we can predict the mean RT for a set of stimuli and three estimated parameters: the adjustment $\beta$, a decision criterion (A), and a residual time ($T_0$):

$$RT = \frac{A}{\beta p} \tanh(Ap) + T_0 \tag{3}$$

where a difference ($\delta$, 0.05 in the current experiment) results in a perceived difference ($p$) given by Fechner's law:

$$p = \beta \log\left(\frac{C + \delta/2}{C - \delta/2}\right) \tag{4}$$

This model then states that the decision easiness (i.e., drift rate, $\beta p$) is determined by the perceived difference between two stimuli $p$. The proportion of correct responses ($P_C$), given the definition of the proportional rate diffusion model (*Palmer et al., 2005*) is then given by

$$P_c = \frac{1}{1 + e^{-2Ap}} \tag{5}$$

The model thus predicts that, for a constant physical difference $\delta$ between stimulus, the RT will increase with the overall contrast values (*Figure 1*) while the likelihood of being correct will decrease.

All these models were fitted per participant using the trust region reflective algorithm implemented in *scipy* (v1.15.2) (*Virtanen et al., 2020*). Models were fitted individually; predictions presented in the paper were aggregated across individual predictions. The estimation of the exponent and slope of Piéron's law was constrained to be negative according to the theoretical formulation. The Fechner diffusion model was constrained based on the parameters observed for similar models (*Tran et al., 2020*). The cross-validation was performed by estimating the models with one contrast value left out and evaluated by computing the rootsquare of the mean-squared prediction error between the left-out contrast and the predictions of the model.

## Acknowledgements

We thank Kenneth Müller for his help in coding the task and collecting the data and Samson Chota for his suggestions on the analyses presented in the paper. We thank Leon Kenemans, Chris Klink, Ben Harvey, Mathieu Servant, and Michael Nunez for the discussion on the present findings. Finally, GW wishes to thank Anna Montagnini and Frédéric Chavane for the informal discussion that ultimately led to this paper. No AI-assisted technologies were used to write the paper or the associated code.

## Additional information

### Funding

| Funder | Grant reference number | Author |
|---|---|---|
| Horizon Europe Marie Sklodowska-Curie Actions | 10.3030/101066503 | Gabriel Weindel Leendert van Maanen |
| Air Force Research Laboratory | OARD FA8655-22-1-7003 | Jelmer P Borst Leendert van Maanen |

The funders had no role in study design, data collection and interpretation, or the decision to submit the work for publication.

### Author contributions

Gabriel Weindel, Conceptualization, Data curation, Software, Formal analysis, Funding acquisition, Investigation, Visualization, Methodology, Writing – original draft; Jelmer P Borst, Conceptualization, Resources, Data curation, Supervision, Funding acquisition, Validation, Investigation, Methodology, Writing – review and editing; Leendert van Maanen, Conceptualization, Supervision, Funding acquisition, Validation, Investigation, Methodology, Project administration, Writing – review and editing

### Author ORCIDs

Gabriel Weindel ⓘ https://orcid.org/0000-0002-7592-1686
Jelmer P Borst ⓘ https://orcid.org/0000-0002-4493-8223
Leendert van Maanen ⓘ http://orcid.org/0000-0001-9120-1075

### Ethics

Human subjects: The experiment was conducted in agreement with the declaration of Helsinki. The participants signed an informed consent form to participate in the study after having read an information brochure on the purpose of the research. The protocol, study material and data sharing policy were validated by the CETO ethical committee of the University of Groningen (ID 94056673).

Reviewer #1 (Public review): https://doi.org/10.7554/eLife.108049.3.sa1
Reviewer #2 (Public review): https://doi.org/10.7554/eLife.108049.3.sa2
Reviewer #3 (Public review): https://doi.org/10.7554/eLife.108049.3.sa3
Author response https://doi.org/10.7554/eLife.108049.3.sa4

# Additional files

## Supplementary files
MDAR checklist

## Data availability
The data and code associated with this manuscript can be found on the associated OSF (https://osf.io/j6xuv) and Github (https://github.com/GWeindel/decision-times, copy archived at *Weindel, 2025*).

The following dataset was generated:

| Author(s) | Year | Dataset title | Dataset URL | Database and Identifier |
|---|---|---|---|---|
| Weindel G, Borst JP, van Maanen L | 2025 | Decision-making components and times revealed by the single-trial electro-encephalogram | https://osf.io/j6xuv | Open Science Framework, j6xuv |

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

# Appendix 1

## Assuming half-sines when the signal is a ramp

Based on two theories of ERP generation (*Makeig, 2002*; *Başar, 1980*), the HMP method assumes that searching for multivariate half-sines in the EEG signal is relevant to capture significant changes in cognitive processing. In the case of the centro-parietal positivity, however, it is believed that this potential arises from ramping activity at the single-trial level (*O'Connell et al., 2012*). Nevertheless, the HMP analyses presented in the main text show that we were able to find this ramp, therefore suggesting that the method can capture alternative generating signals. In this Appendix, to formally confirm that the HMP method also detects an event if it was comprised of a ramp-like signal, we simulated data congruent with this hypothesis vs. the half-sine assumption.

To do so, we first downsampled the signal to 100 Hz for computational reasons and reestimated an HMP model as in the main results but with a 50 ms event width due to the new sampling rate. In this downsampled dataset, the optimal solution is three HMP events (*Appendix 1—figure 1A*) instead of five as in the main text. We then used the downsampled data structure without any real data and added, at the single-trial most-likely times, the pattern of each event. To add the pattern to the 0 signal, we computed the outer product between the patterns' expected activation (either a ramp or a half-sine) and the values of the electrodes at the trial-by-trial most likely time of each event based on the estimates from the real data. The two first events were always simulated as half-sines: the activation pattern was defined as the first five samples of a full 10 Hz sine wave sampled at 100 Hz with a phase of 0 (i.e., 50 ms pulse-like activity). The last event was simulated as either a half-sine with the same characteristics as the first two or a ramp. The ramp pattern was defined per trial by generating 100,000 cumulative sums of Gaussian noise with both mean and standard deviation set to 0.1 over 2000 time steps. These cumulative sums form simulated accumulation traces and thus the ramp expected in decision-related signals. For each simulated accumulation trace, we recorded the time step at which a decision threshold was crossed by monitoring the first time the absolute cumulative sum exceeded the value 1 (traces that reached a threshold of -1 were inverted). For each simulated trial, to avoid a too high impact of noise, we averaged the 10 traces with the closest time to the actual time between the second event and the response. This ramp-like pattern was then added between the N200 event and the response for each trial. After having imputed the three patterns to each trial at their corresponding times, we added noise based on the variance–covariance matrix between electrodes observed in the real data, computed during the 200 ms baseline of the epochs, using a multivariate normal and an infinite impulse response filter (with 0.2, –0.2, and 0.04 as denominator coefficients as done in *Weindel et al., 2024*) as implemented in scipy (v1.15.2; *Virtanen et al., 2020*). We then estimated the same HMP model as for the downsampled data in *Appendix 1—figure 1A*.

*Appendix 1—figure 1B* illustrates the topography of the last event for the simulated ramp signal (left panel) vs. a simulated half-sine (right panel). In both cases, the last topography detected by HMP is very close to the last one estimated on the downsampled real data (*Figure 1A*, top). Representing the event's match between the data and the samples preceding the peak of the event (as in *Figure 5C*) shows that the build-up of the event clearly indicates whether the generating single-trial pattern is a ramp or a half-sine. Thus, in this Appendix, we show that, irrespective of the generating signal being a ramp or a half-sine, if this signal is congruent with HMPs' assumptions of sequentiality and repetition across trials (see Section 3 of *Weindel et al., 2024* for a discussion), then this event will be detected. We also show that the build-up of this event can be used to infer the single-trial generating signal.

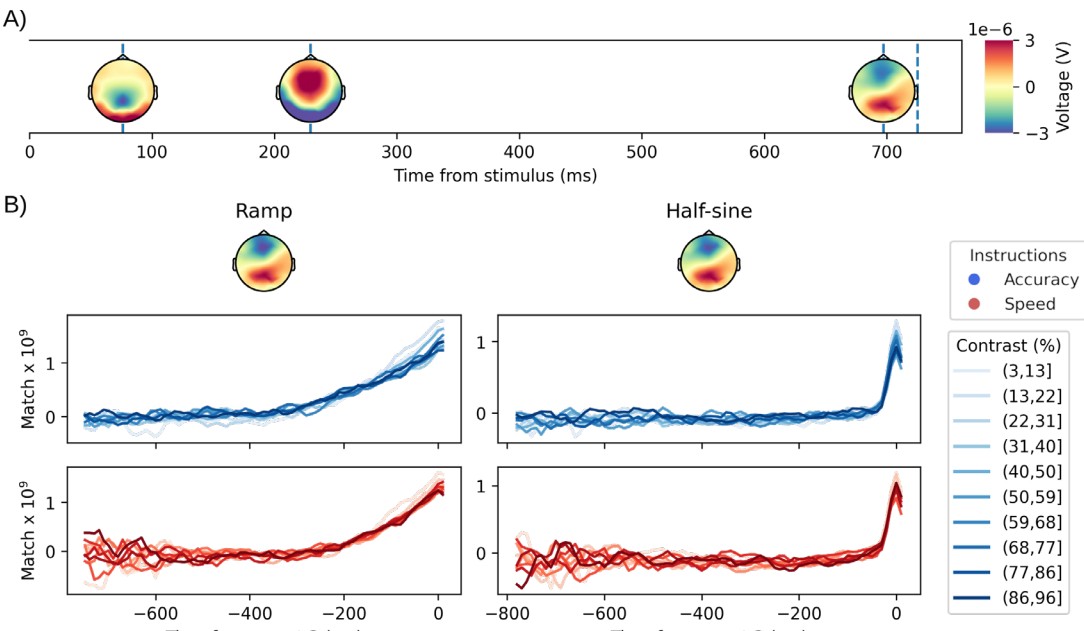

**Appendix 1—figure 1.** Simulating a ramp or a half-sine and expecting a half-sine. (**A**) Representation of the averaged times and topographies for the three events detected by HMP on the downsampled real data. (**B**) Trial and participant-averaged topography (top panels) and time courses (bottom panels) of the electrodes matched to the third HMP event for the ramp simulation (left panels) vs. the half-sine simulation (right panels). As in the main text, the trials were first centered on the peak of the third event before computing the average ERP for the different contrast values binned into 10 bins.

