## [Editor Report · eLife Assessment]

Weindel et al. examine behavioral and EEG data in an innovative contrast comparison paradigm where they vary mean contrast widely while keeping contrast difference constant. As intended, this allowed an elegant decomposition of processing stages: while sensory encoding shortened with increasing contrast in keeping with Pieron’s law, the period of decision formation lengthened, in keeping with Fechner’s law, which was applied to drift rates in a diffusion model of that period. This is an **important** demonstration of how these two laws apply in concert, to two distinct processing levels, and the multivariate topography parsing, mixed effect models, and diffusion models are **convincing**.

---

## [Referee Report · Reviewer #1 (Public review)]

This study uses a new 'hidden multivariate pattern method' to parse in time and space the neural events intervening between stimulus and response in an immediately-reported perceptual decision, and use the resultant neural event timing information to show quite convincingly that Pieron's and Fechner's laws can apply in concert at distinct processing levels.

They designed a clever contrast comparison paradigm in which the contrast difference is kept constant while widely manipulating mean contrast, so that sensory encoding of the overall stimulus would be boosted with increasing mean contrast, whereas decision difficulty and hence duration would increase. With this, they found that the time intervening between early sensory-evoked components, up to an 'N200'-type component associated with launching the decision process, varies inversely with contrast according to Pieron's law. Meanwhile, the time intervals running up to neural events peaking near the time of response, consistent with decision termination, increases with contrast, fitting Fechner's law. Further, a diffusion model whose drift rates are scaled by Fechner's law, fit to RT, predicts the observed proportion of correct responses very well.

In the process of review and revision it was highlighted that presumably the full sequence of neural events intervening between stimulus and response is massively task dependent, but;

(1) The method is intended to capture all key components that specifically covary with RT, as opposed to each and every component in general, and

(2) The main conclusions of the study mentioned above do not change whether the method is set up to track three neural events, or five, as was done in the final analysis.

The propensity for topographic parsing algorithms to potentially lump-together distinct processes that partially co-evolve was acknowledged, but a key clarification in review was that even though the method entails a specification of neural event duration - which was changed from 50 to 25 ms - the success of the method is not strongly contingent on the actual underlying neural events in question having that very duration - indeed, the components extracted using that short template duration can be observed to evolve over a longer time frame associated with the Fechner diffusion process.

Notably, standard average event-related potential analysis was able to show expected amplitude effects - where sensory signals increased with contrast but decision signals decreased - but assessment of the by-trial distribution of their timings was grealy aided by the HMP method.

One of the stages of processing implicated in the parsing analysis was linked to attention orientation, and the authors speculate on whether this might reflect a spatially-selective deployment of attention or a resource allocation, but sensibly refrain from speculating too far since the focus here was on the sensory and decision process durations and their respective adherence to Pieron and Fechner's laws.

---

## [Referee Report · Reviewer #2 (Public review)]

Summary:

The authors decomposed response times into component processes and manipulated the duration of these processes in opposing directions by varying contrast, and overall by manipulating speed-accuracy tradeoffs. They identify different processes and their durations by identifying neural states in time and validate their functional significance by showing that their properties vary selectively as expected with predicted effects of the contrast manipulation. They identify 4 processes: stimulus encoding, attention orienting, decision and motor execution. These map onto 5 classical event related potentials. The decision-making component matched the CPP and its properties varied with contrast and predicted decision-accuracy.

Strengths:

The design of the experiment is remarkable and offers crucial insights. The analyses techniques are beyond-state-of-the art and the analyses are well motivated and offer clear insights.

Weaknesses:

The number of identified events depends on the parameter setting of the analysis. While the authors discuss weaknesses of the approach this needs to be made explicit as well. It is also unclear to what extent topographies map onto processes since e.g., different combinations of sources can lead to the same scalp topography.

---

## [Referee Report · Reviewer #3 (Public review)]

Summary:

In this manuscript the authors examine the processing stages involved in perceptual decision-making using a new approach to analysing EEG data, combined with a critical stimulus manipulation. This new EEG analysis method enables single-trial estimates of the timing and amplitude of transient changes in EEG time-series recurrent across trials in a behavioural task. The authors find evidence for five events between stimulus onset and the response in a two-spatial-interval visual discrimination task. By analysing the timing and amplitude of these events in relation to behaviour and the stimulus manipulation, the authors interpret these events as related to separable processing stages for stimulus encoding (first two events), attention orientation (second event), motor planning (fourth event) and decision (deliberation, final event). This is largely consistent with previous findings from both event-related potentials (across trials) and single-trial estimates using decoding techniques and neural network approaches. However, by taking a data-driven approach (as opposed to theory-driven decoding analyses) a more nuanced picture emerges: there are several stimulus encoding steps which may contribute differently to behaviour, and decision processes extend beyond the planning of the motor response.

Strengths:

This work is not only important for the conceptual advance, but also in promoting this new analysis technique, which will likely prove useful in future research. For the broader picture, this work is an excellent example of the utility of neural measures for mental chronometry.

Weaknesses:

Though beyond the scope of this manuscript, these results should be considered within the broader decision-making literature, where task or domain-specific processes may not generalise (for example, in value-based decision-making).

---

## [Author Response]

The following is the authors’ response to the original reviews.

**Reviewer #1 (Public review):**
From my reading, this study aimed to achieve two things:(1) A neurally-informed account of how Pieron's and Fechner's laws can apply in concert at distinct processing levels.(2) A comprehensive map in time and space of all neural events intervening between stimulus and response in an immediately-reported perceptual decision.I believe that the authors achieved the first point, mainly owing to a clever contrast comparison paradigm, but with good help also from a new topographic parsing algorithm they created. With this, they found that the time intervening between an early initial sensory evoked potential and an "N2" type process associated with launching the decision process varies inversely with contrast according to Pieron's law. Meanwhile, the interval from that second event up to a neural event peaking just before response increases with contrast, fitting Fechner's law, and a very nice finding is that a diffusion model whose drift rates are scaled by Fechner's law, fit to RT, predicts the observed proportion of correct responses very well. These are all strengths of the study.

We thank the reviewer for their comments that added context to the events we detected in relation to previous findings. We also believe that the change in the HMP algorithm suggested by the reviewer improved the precision of our analyses and the manuscript. We respond to the reviewer’s specific comments below.

(1) The second, generally stated aim above is, in the opinion of this reviewer, unconvincing and ill-defined. Presumably, the full sequence of neural events is massively task-dependent, and surely it is more in number than just three. Even the sensory evoked potential typically observed for average ERPs, even for passive viewing, would include a series of 3 or more components - C1, P1, N1, etc. So are some events being missed? Perhaps the authors are identifying key events that impressively demarcate Pieron- and Fechner-adherent sections of the RT, but they might want to temper the claim that they are finding ALL events. In addition, the propensity for topographic parsing algorithms to potentially lump together distinct processes that partially co-evolve should be acknowledged.

We agree with the reviewer that the topographical solutions found by HMP will be dependent on the task and the quality and type of data. We address this point in the last section of the discussion (see also response to R3.5). We would also like to add that the events detected by HMP are, by construction, those that contribute to the RT and not necessarily all ERPs elicited by a stimulus.

In addition to the new last section of the discussion we also make these points clear in the revised manuscript at the discussion start:

“By modeling the recorded single-trial EEG signal between stimulus onset and response as a sequence of multivariate events with varying by-trial peak times, we aimed to detect recurrent events that contribute to the duration of the reaction time in the present perceptual decision-making task”.

Regarding the typical visual ERPs, in response to this comment but also comments R1.2, R1.3 and R2.1, we aimed for a more precise description of the topographies and thus reduced the width of the HMP expected events to 25ms. This ensures that we do not miss events shorter than the initial expectations of 50ms (see Appendix B of Weindel et al., 2024 and also response to R1.3). This new estimation provides evidence for at least two of the visual ERPs that, based on their timings and topographies (in relation with the spatial frequency of the stimulus), we interpret as the N40 and the P100 (see response to R1.5 for the justification of this categorization). We provide a description and justification of the interpretations in the result section “Five trial-recurrent sequential events occur in the EEG during decisions” and the discussion section “Visual encoding time”.

(2) To take a salient example, the last neural event seems to blend the centroparietal positivity with a more frontal midline negativity, some of which would capture the CNV and some motor-execution related components that are more tightly time-locked to, of course, the response. If the authors plotted the traditional single-electrode ERP at the frontal focus and centroparietal focus separately, they are likely to see very different dynamics and contrast- and SAT-dependency. What does this mean for the validity of the multivariate method? If two or more components are being lumped into one neural event, wouldn't it mean that properties of one (e.g., frontal burstiness at response) are being misattributed to the other (centroparietal signal that also peaks but less sharply at response)?

Using the new HMP parameterization described above we show that the reviewer's intuition was correct. Using an expected pattern duration of 25ms the last event in the original manuscript splits in two events. The before-last event, now referred to the lateralized readiness potential (LRP) presents a strong lateralization (Figure 3) with an increased negativity over the motor cortex contralateral to the right hand. The effect of contrast is mostly on the last event that we interpret as the CPP (Figure 5). Despite the improved precision of the topographies of the identified events, it is however to be noted that some components will overlap. If the LRP is generated when a certain amount of evidence is accumulated (e.g. that the CPP crosses a certain value) then a time-based topography will necessarily include that CPP activity in addition to the lateralized potential. We discuss this in the section “Motor execution” of the discussion:

“Adding the abrupt onset of this potential, we believe that this event is the start of motor execution, engaged after a certain amount of evidence. The evidence for this interpretation is manifest in the fact that the event's topography shares some activity with the CPP event that follows, an expected result if the LRP is triggered at a certain amount of evidence, indexed by the CPP”.

(3) Also related to the method, why must the neural events all be 50 ms wide, and what happens if that is changed? Is it realistic that these neural events would be the same duration on every trial, even if their duration was a free parameter? This might be reasonable for sensory and motor components, but unlikely for cognitive.

The HMP method is sensitive to the event's duration as shown in the manuscript about the method (Appendix B of Weindel et al., 2024). Nevertheless as long as the topography in the real data is longer than the expected one it shouldn't be missed (i.e. same goes for by-trial variations in the event width). For this reason we halved the expected event width of 50ms (introduced by the original HsMM-MVPA paper by Anderson and colleagues) in the revision. This new estimation with 25ms thus is much less likely to miss events as evidenced by the new visual and motor events. In the revised manuscript this is addressed at the start of the Results section:

“Contrary to previous applications (Anderson et al.,2016; Berberyan et al., 2021; Zhang et al., 2018; Krause et al., 2024) we assumed that the multivariate pattern was represented by a 25ms half-sine as our previous research showed that a shorter expected pattern width increases the likelihood of detecting cognitive events (see Appendix B of Weindel et al., 2024)”.

Regarding the event width as a free parameter this is both technically and statistically difficult to implement as the amount of computing capacity, flexibility and trade-offs among the HMP parameters would, given the current implementation, render the model unfit for most computers and statistically unidentifiable.

(4) In general, I wonder about the analytic advantage of the parsing method - the paradigm itself is so well-designed that the story may be clear from standard average event-related potential analysis, and this might sidestep the doubts around whether the algorithm is correctly parsing all neural events.

Average ERP analysis suffers from an impossibility to differentiate between an effect of an experimental factor on the amplitude vs. on the timing of the underlying components (Luck, 2005). Furthermore the overlap of components across trials bluries the distinction between them. For both reasons we would not be able to reach the same level of certainty and precision using ERP analyses. Furthermore the relatively low number of trials per experimental cell (contrast level X SAT X participant = 6 trials) makes the analyses hard to perform on ERP which typically require more trials per modality. From the reviewer’s comment we understand that this point was not clear. We therefore discuss this in the revision, Section “Functional interpretation of the events” of the results:

“Nevertheless identifying neural dynamics on these ERPs centered on stimulus is complicated by the time variation of the underlying single-trial events (see probabilities displayed in Figure 3 for an illustration and Burle et al., 2008, for a discussion). The likely impact of contrast on both amplitude and time on the underlying single-trial event does not allow one to interpret the average ERP traces as showing an effect in one or the other dimension without strong assumptions (Luck, 2005)”.

(5) In particular, would the authors consider plotting CPP waveforms in the traditional way, across contrast levels? The elegant design is such that the C1 component (which has similar topography) will show up negative and early, giving way to the CPP, and these two components will show opposite amplitude variations (not just temporal intervals as is this paper's main focus), because the brighter the two gratings, the stronger the aggregate early sensory response but the weaker the decision evidence due to Fechner. I believe this would provide a simple, helpful corroborating analysis to back up the main functional interpretation in the paper.

We agree with the suggestion and have introduced the representation on top of Figure 5 for sets of three electrodes in the occipital, posterior and frontal regions. The new panels clearly show an inversion of the contrast effect dependent on the time and locus of the electrodes. We discuss this in Section “Functional interpretation of the events” of the results:

“This representation shows that there is an inversion of the contrast effect with higher contrasts having a higher amplitude on the electrodes associated with visual potentials in the first couple of deciseconds (left panel of Figure 5A) while parietal and frontal electrodes shows a higher amplitude for lower contrasts in later portions of the ERPs (middle and right panel of Figure 5A)”.

To us, this crucially shows that we cannot achieve the same decomposition using traditional ERP analyses. In these plots it appears that while, as described by the reviewer, there is an inversion, the timing and amplitude of the changes due to contrast can hardly be interpreted.

(6) The first component is picking up on the C1 component (which is negative for these stimulus locations), not a "P100". Please consult any visual evoked potential study (e.g., Luck, Hillyard, etc). It is unexpected that this does not vary in latency with contrast - see, for example. Gebodh et al (2017, Brain Topography) - and there is little discussion of this. Could it be that nonlinear trends were not correctly tested for?

We disagree with the reviewer on the interpretation of the ERP. The timing of the detected component is later than the one usually associated with a C1. Furthermore the central display does not create optimal conditions to detect a C1

We do agree that the topography raises the confusion but we believe that this is due to the spatial frequency of the stimulus that generates a high posterior positivity (see references in the following extract). The new HMP solution also now happens to show an effect of contrast on the P100 latencies, we believe this is due to the increased precision in the time location of the component. We discuss this in the “Visual encoding time” section of the discussion:

“The following event, the P100, is expressed around 70ms after the N40, its topography is congruent with reports for stimuli with low spatial frequencies as used in the current study (Kenemans et al., 2002, 2000; Proverbio et al., 1996). The timing of this P100 component is changed by the contrast of the stimulus in the direction expected by the Piéron law (Figure 4A)”.

(7) There is very little analysis or discussion of the second stage linked to attention orientation - what would the role of attention orientation be in this task? Is it spatial attention directed to the higher contrast grating (and if so, should it lateralise accordingly?), or is it more of an alerting function the authors have in mind here?

We agree that we were not specific enough on the interpretation of this attention stage. We now discuss our hypothesis in the section “Attention orientation” of the discussion:

“We do however observe an asymmetry in the topographical map Figure 3. This asymmetry might point to an attentional bias with participants (or at least some participants) allocating attention to one side over the other in the same way as the N2pc component (Luck and Hillyard, 1994, [40]). Based on this collection of observations, we conclude that this third event represents an attention orientation process. In line with the finding of Philiastides et al. (2006), this attention orientation event might also relate to the allocation of resources. Other designs varying the expected cognitive load or spatial attention could help in further interpreting the functional role of this third event”.

We would like to add that it is unlikely that the asymmetry we mention in the discussion cannot stem from the redirection towards higher contrast as the experimental design balanced the side of presentation. We therefore believe that this is a behavioral bias rather than a bias toward the highest contrast stimulus as suggested by the reviewer. We hope that, while more could be tested and discussed, this discussion is sufficient given the current manuscript's goal.

**Reviewer #2 (Public review):**
Summary:The authors decomposed response times into component processes and manipulated the duration of these processes in opposing directions by varying contrast, and overall by manipulating speed-accuracy tradeoffs. They identify different processes and their durations by identifying neural states in time and validate their functional significance by showing that their properties vary selectively as expected with the predicted effects of the contrast manipulation. They identify 3 processes: stimulus encoding, attention orienting, and decision. These map onto classical event-related potentials. The decision-making component matched the CPP, and its properties varied with contrast and predicted decision-accuracy, while also exhibiting a burst not characteristic of evidence accumulation.Strengths:The design of the experiment is remarkable and offers crucial insights. The analysis techniques are beyond state-of-the-art, and the analyses are well motivated and offer clear insights.Weaknesses:It is not clear to me that the results confirm that there are only 3 processes, since e.g., motor preparation and execution were not captured. While the authors discuss this, this is a clear weakness of the approach, as other components may also have been missed. It is also unclear to what extent topographies map onto processes, since, e.g., different combinations of sources can lead to the same scalp topography.

We thank the reviewer for their kind words and for the attention they brought on the question of the missing motor preparation event. In light of this comment (and also R1.1, R3.3) the revised manuscript uses a finer grained approach for the multivariate event detection. This preciser estimation comes from the use of a shorter expected pattern in which the initial expectation of a 50ms half-sine was halved, therefore ensuring that we do not miss events shorter than the initial expectations (see Appendix B of Weindel et al., 2024 and also response to R1.3). In the new solution the motor component that the reviewer expected is found as evidenced by the topography of the event, its lateralization and a time-to-response congruent with a response execution event. This is now described in the section “Motor execution” of the revised manuscript:

“The before last event, identified as the LRP, shows a strong hemispheric asymmetry congruent with a right hand response. The peak of this event is approximately 100 ms before the response which is congruent with reports that the LRP peaks at the onset of electromyographical activity in the effector muscle (Burle et al., 2004), typically happening 100ms before the response in such decision-making tasks (Weindel et al., 2021). Furthermore, while its peak time is dependent on contrast, its expression in the EEG is less clearly related to the contrast manipulation than the following CPP event”.

**Reviewer #3 (Public review):**
Summary:In this manuscript, the authors examine the processing stages involved in perceptual decision-making using a new approach to analysing EEG data, combined with a critical stimulus manipulation. This new EEG analysis method enables single-trial estimates of the timing and amplitude of transient changes in EEG time-series, recurrent across trials in a behavioural task. The authors find evidence for three events between stimulus onset and the response in a two-spatial-interval visual discrimination task. By analysing the timing and amplitude of these events in relation to behaviour and the stimulus manipulation, the authors interpret these events as related to separable processing stages for stimulus encoding, attention orientation, and decision (deliberation). This is largely consistent with previous findings from both event-related potentials (across trials) and single-trial estimates using decoding techniques and neural network approaches.Strengths:This work is not only important for the conceptual advance, but also in promoting this new analysis technique, which will likely prove useful in future research. For the broader picture, this work is an excellent example of the utility of neural measures for mental chronometry.

We appreciate the very positive review and thank the reviewer for pointing out important weaknesses in our original manuscript and also providing resources to address them in the recommendations to authors. Below we comment on each identified weakness and how we addressed them.

Weaknesses:(1) The manuscript would benefit from some conceptual clarifications, which are important for readers to understand this manuscript as a stand-alone work. This includes clearer definitions of Piéron's and Fechner's laws, and a fuller description of the EEG analysis technique.

We agree that the description of both laws were insufficient, we therefore added the following text in the last paragraph of the introduction:

“Piéron’s law predicts that the time to perceive the two stimuli (and thus the choice situation) should follow a negative power law with the stimulus intensity (Figure 1, green curve). In contradistinction, Fechner’s law states that the perceived difference between the two patches follows the logarithm of the absolute contrast of the two patches (Figure 1, yellow curve). As the task of our participants is to judge the contrast difference, Piéron’s law should predict the time at which the comparison starts (i.e. the stimuli become perceptible), while Fechner’s law should implement the comparison, and thus decision, difficulty”.

Regarding the EEG analysis technique we added a few elements at the start of the result:

“The hidden multivariate pattern model (HMP) implemented assumed that a task-related multivariate pattern event is represented by a half-sine whose timing varies from trial to trial based on a gamma distribution with a shape parameter of 2 and a scale, controlling the average latency of the event, free-to-vary per event (Weindel et al., 2024)”.

We also made the technique clearer at the start of the discussion:

“By modeling the recorded single-trial EEG signal between stimulus onset and response as a sequence of multivariate events with varying by-trial peak times, we aimed to detect recurrent events that contribute to the duration of the reaction time in the present perceptual decision-making task. In addition to the number of events, using this hidden multivariate pattern approach (Weindel et al., 2024) we estimated the trial-by-trial probability of each event’s peak, therefore accessing at which time sample each event was the most likely to occur”.

Additionally, we added a proper description in the method section (see the new first paragraph of the “Hidden multivariate pattern” subsection).

(2) The manuscript, broadly, but the introduction especially, may be improved by clearly delineating the multiple aims of this project: examining the processes for decision-making, obtaining single-trial estimates of meaningful EEG-events, and whether central parietal positivity reflects ramping activity or steps averaged across trials.

For the sake of clarity we removed the question of the ramping activity vs steps in the introduction and focused on the processes in decision-making and their single-trial measurement as this is the main topic of the paper. Furthermore the references provided by the reviewer allowed us to write a more comprehensive review of previous studies and how the current study is in line with those. These changes are mainly manifested in these new sentences:

“As an example Philiastides et al. (2006) used a classifier on the EEG activity of several conditions to show that the strength of an early EEG component was proportional to the strength of the stimulus while a later component was related to decision difficulty and behavioral performance (see also Salvador et al., 2022; Philiastides and Sajda, 2006). Furthermore the authors interpreted that a third EEG component was indicative of the resource allocated to the upcoming decision given the perceived decision difficulty. In their study, they showed that it is possible to use single-trial information to separate cognitive processes within decision-making. Nevertheless, their method requires a decoding approach, which requires separate classifiers for each component of interest and restrains the detection of the components to those with decodable discriminating features (e.g. stimuli with strong neural generators such as face stimuli, see Philiastides et al., 2006)”.

(3) A fuller discussion of the limitations of the work, in particular, the absence of motor contributions to reaction time, would also be appreciated.

As laid out in responses to comments R1.1 and R2 the new estimates now include evidence for a motor preparation component. We discuss this in the new “motor execution” paragraph in the discussion section. Additionally we discuss the limitation of the study and the method in the two last paragraphs of the discussion (in the new Section “Generalization and limitation”).

(4) At times, the novelty of the work is perhaps overstated. Rather, readers may appreciate a more comprehensive discussion of the distinctions between the current work and previous techniques to gauge single-trial estimates of decision-related activity, as well as previous findings concerning distinct processing stages in decision-making. Moreover, a discussion of how the events described in this study might generalise to different decision-making tasks in different contexts (for example, in auditory perception, or even value-based decision-making) would also be appreciated.

We agree that the original text could be read as overstating. In addition to the changes linked to R3.2 we also now discuss the link with the previous studies in the before-last paragraph of the discussion before the conclusion in the new “Generalization and limitations” section:

“The present study showed what cognitive processes are contributing to the reaction time and estimated single-trial times of these processes for this specific perceptual decision-making task. The identified processes and topographies ought to be dependent on the task and even the stimuli (e.g. sensory events will change with the sensory modality). More complex designs might generate a higher number of cognitive processes (e.g. memory retrieval from a cue, Anderson et al., 2016) and so could more natural stimuli which might trigger other processes in the EEG (e.g. appraisal vs. choice as shown by Frömer et al., 2024). Nevertheless, the observation of early sensory vs. late decision EEG components is likely to generalize across many stimuli and tasks as it has been observed in other designs and methods (Philiastides et al., 2006; Salvador et al., 2022). To these studies we add that we can evaluate the trial-level contribution, as already done for specific processes (e.g. Si et al., 2020; Sturm et al., 2016), for the collection of events detected in the current study”.

**Reviewing Editor Comments:**

As you will see, all three reviewers agree that the paper makes a valuable contribution and has many strengths. You will also see that they have provided a range of constructive comments highlighting potential issues with the interpretation of the outcomes of your signal decomposition method. In particular, all three reviewers point out that your results do not identify separate motor preparation signals, which we know must be operating on this type of task. The reviewers suggest further discussion of this issue and the potential limitations of your analysis approach, as well as suggesting some additional analyses that could be run to explore this further. While making these changes would undoubtedly enhance the paper and the final public reviews, I should note that my sense is that they are unlikely to change the reviewers' ratings of the significance of the findings and the strength of evidence in the final eLife assessment

**Reviewer #1 (Recommendations for the authors):**
(1) Abstract: "choice onset" is ill-defined and not the label most would give the start of the RT interval. Do you mean stimulus onset?

We replaced with "choice onset" with "stimulus onset" in the abstract

(2) Similarly "choice elements" in the introduction seem to refer to sensory attributes/objects being decided about?

We replaced "choice-elements" with "choice-relevant features of the stimuli"

(3) "how the RT emerges from these putative components" - it would be helpful to specify more what level of answer you're looking for, as one could simply answer "when they're done."

We replaced with "how the variability in RTs emerges from these putative components"

(4) Line 61-62: I'm not sure this is a fully correct characterisation of Frömer et al. It was not similar in invoking a step function - it did not invoke any particular mechanism or function, and in that respect does not compare well to Latimer et al. Also, I believe it was the overlap of stimulus-locked components, not response-locked, that they argued could falsely generate accumulator-like buildup in the response-locked ERP.

We indeed wrongly described Frömer et al. The sentence is now "In human EEG data, the classical observation of a slowly evolving centro-parietal positivity, scaling with evidence accumulation, was suggested to result from the overlap of time-varying stimulus-related activity in the response-locked event related potential"

(5) Line 78: Should this be single-trial *latency*?

This referred to location in time but we agree that the term is confusing and thus replaced it with latencies.

(6) The caption of Figure 1 should state what is meant by the y-axis "time"

We added the sentence "The y-axis refers the time predicted by each law given a contrast value (x-axis) and the chosen set of parameters." in the caption of Figure 1

(7) Line 107: Is this the correct description of Fechner's law? If the perceived difference follows the log of the physical difference, then a constant physical difference should mean a constant perceived difference. Perhaps a typo here.

This was indeed a typo we replaced the corresponding part of the sentence with "the perceived difference between the two patches follows the logarithm of the absolute contrast of the two patches"

(8) Line 128: By scale, do you mean magnitude/amplitude?

No, this refers to the parameter of a gamma distribution. To clarify we edited the sentence: "based on a gamma distribution with a shape parameter of 2 and a scale parameter, controlling the average latency of the event, free-to-vary per event"

(9) The caption of Figure 3 is insufficient to make sense of the top panel. What does the inter-event interval mean, and why is it important to show? What is the "response" event?

We agree that the top panel was insufficiently described. To keep the length of the paper short and because of the relatively low amount of information provided by these panels we replaced them for a figure only showing the average topographies as well as the asymmetry tests for each event.

(10) Figure 4: caption should say what the top vs bottom row represents (presumably, accuracy vs speed emphasis?), and what the individual dots represent, given the caption says these are "trial and participant averaged". A legend should be provided for the rightmost panels.

We agree and therefore edited Figure 4. The beginning of the caption mentioned by the reviewer now reads: “(A) The panels represent the average duration between events for each contrast level, averaged across participants and trials (stimulus and response respectively as first and last events) for accuracy (top) and speed instructions (bottom).”. Additionally we added legends for the SAT instructions and the model fits.

(11) Line 189: argued for a decision-making role of what?

Stafford and Gurney (2004) proposed that Pieron’s law could reflect a non-linear transformation from sensory input to action outcomes, which they argued reflected a response mechanism. We (Van Maanen et al., 2012) specified this result by showing that a Bayesian Observer Model in which evidence for two alternative options was accumulated following Bayes Rule indeed predicted a power relation between the difference in sensory input of the two alternatives, and mean RT. However, the current data suggest that such an explanation cannot be the full story, as also noted by R3. To clarify this point we replaced the comment by the following sentence:

“Note that this observation is not necessarily incongruent with theoretical work that argued that Piéron’s law could also be a result of a response selection mechanism (Stafford and Gurney, 2004; Van Maanen et al., 2012; Palmer et al., 2005). It could be that differences in stimulus intensity between the two options also contribute to a Piéron-like relationship in the later intervals, that is convoluted with Fechner’s law (see Donkin and Van Maanen, 2014 for a similar argument). Unfortunately, our data do not allow us to discriminate between a pure logarithmic growth function and one that is mediated by a decreasing power function”.

(12) Table 2: There is an SAT effect even on the first interval, which is quite remarkable and could be discussed more - does this mean that the C1 component occurs earlier under speed pressure? This would be the first such finding.

The original event we qualified as a P100 was sensitive to SAT but the earliest event is now the N40 and isn’t statistically sensitive to speed pressure in this data. We believe that the fact that the P100 is still sensitive to SAT is not a surprise and therefore do not outline it.

(13) Line 221: "decrease of activation when contrast (and thus difficulty) increases" - is this shown somewhere in the paper?

The whole section for this analysis was rewritten (see comment below)

(14) I find the analysis of Figure 5 interesting, but the interpretation odd. What is found is that the peak of the decision signal aligns with the response, consistent with previous work, but the authors choose to interpret this as the decision signal "occurring as a short-lived burst." Where is the quantitative analysis of its duration across trials? It can at least be visually appraised in the surface plot, and this shows that the signal has a stimulus-locked onset and, apart from the slowest RTs, remains present and for the most part building, until response. What about this is burst-like? A peak is not a burst.

This was the residue of a previous version of the paper where an analysis reported that no evidence accumulation trace was found. But after proper simulations this analysis turned out to be false because of a poor statistical test. Thus we removed this paragraph in the revised manuscript and Figure 5 has now been extended to include surface plots for all the events.

**Reviewer #2 (Recommendations for the authors):**
Overall, I really enjoyed reading this paper. However, in some places the approach is a bit opaque or the results are difficult to follow. As I read the paper, I noted:Did you do a simple DDM, or did you do a collapsing bound for speed?

The fitted DDM was an adaptation of the proportional rate diffusion model. We make this clearer at the end of the introduction: "Given that Fechner’s law is expected to capture decision difficulty we connected this law to the classical diffusion decision models by replacing the rate of accumulation with Fechner’s law in the proportional rate diffusion model of Palmer et al.(2005).”

It is confusing that the order of intervals in the text doesn't match the order in the table. It might be better to say what events the interval is between rather than assuming that the reader reconstructs.

We agree and adapted the order in both the text and the table. The table is now also more explicit (e.g. RT instead of S-R)

Otherwise, I do wonder to what extent the method is able to differentiate processes that yield similar scalp topographies and find it a bit concerning that no motor component was identified.

We believe that the new version with the LRP/CPP is a demonstration that the method can handle similar topographies. The method can handle events with close topographies as long as they are separate in time, however if they are not sequential to one another the method cannot capture both events. We now discuss this, in relation with the C1/P100 overlap, in the discussion section “Visual encoding time”:

“Nevertheless this event, seemingly overlapping with the P100 even at the trial level (Figure 5C), cannot be recovered by the method we applied. The fact that the P100 was recovered instead of the C1 could indicate that only the timing of the P100 contributes to the RT (see Section 3 of Weindel et al., 2024)”.

And we more generally address the question of overlap in the new section “Generalization and limitation”.

**Reviewer #3 (Recommendations for the authors):**
Major Comments:(1) If we agree on one thing, it is that motor processes contribute to response time. Line 364: "In the case of decision-making, these discrete neural events are visual encoding, attention-orientation, and decision commitment, and their latency make up the reaction time." Does the third event, "decision commitment", capture both central parietal positivity (decision deliberation) and motor components? If so, how can the authors attribute the effects to decision deliberation as opposed to motor preparation?

Thanks to the suggestions also in the public part. This main problem is now addressed as we do capture both a motor component and a decision commitment.

Line 351 suggests that the third event may contain two components.

This was indeed our initial, badly written, hypothesis. Nevertheless the new solution again addresses this problem.

The time series in Figure 6 shows an additional peak that is not evident in the simulated ramp of Appendix 1.

This was probably due to the overlap of both the CPP and the LRP. It is now much clearer that the CPP looks mostly like a ramp while the LRP looks much more like a burst-like/peaked activity. We make this clear in the “Decision event” paragraph of the discussion section:

“Regarding the build-up of this component, the CPP is seen as originating from single-trial ramping EEG activities but other work (Latimer et al., 2015; Zoltowski et al., 2019) have found support for a discrete event at the trial-level. The ERPs on the trial-by-trial centered event in Figure 5 show support for both accounts. As outlined above, the LRP is indeed a short burst-like activity but the build-up of the CPP between high vs low contrast diverges much earlier than its peak”.

Previous analyses (Weindel et al., 2024) found motor-related activity from central parietal topographies close to the response by comparing the difference in single-trial events on left- vs right-hand response trials. The authors suggest at line 315 that the use of only the right hand for responding prevented them from identifying a motor event.The use of only the right hand should have made the event more identifiable because the topography would be consistent across trials (rather than inverting on left vs right hand response trials).

The reviewer is correct, in the original manuscript we didn’t test for lateralization, but the comment of the reviewer gave us the idea to explicitly test for the asymmetry (Figure 3). This test now clearly shows what would be expected for a motor event with a strong negativity over the left motor cortex.

The authors state on line 422 that the EEG data were truncated at the time of the response.Could this have prevented the authors from identifying a motor event that might overlap with the timing of the response?

We thank the reviewer for this suggestion. This would have been a possibility but the problem is that adding samples after the response also adds the post-response processes (error monitoring, button release, stimulus disappearance, etc.). While increasing the samples after the response is definitely something that we need to inspect, we think that the separation we achieved in this revision doesn’t call for this supplementary analysis.

The largest effects of contrast on the third event amplitude appear around the peak as opposed to the ramp. If the peak is caused by the motor component, how does this affect the conclusions that this third event shows a decision-deliberation parietal processes as opposed to a motor process (a number of studies suggest a causal role for motor processes in decision-making e.g. Purcell et al., 2010 Psych Rev; Jun et al., 2021 Nat Neuro; Donner et al., 2009 Curr Bio).

This result now changed and it does look like the peak capturing most of the effect is no longer true. We do however think that there might be some link to theories of motor-related accumulation. We therefore added this to the discussion in the Motor execution section:

“Based on all these observations, it is therefore very likely that this LRP event signs the first passage of a two-step decision process as suggested by recent decision-making models (Servant et al., 2021; Verdonck et al., 2021; Balsdon et al., 2023)”.

I would suggest further investigation into the motor component (perhaps by extending the time window of analysed EEG to a few hundred ms after the response) and at least some discussion of the potential contribution of motor processes, in relation to the previous literature.

We believe that the absence of a motor component is sufficiently addressed in the revised manuscript and in the responses to the other comments.

(2) What do we learn from this work? Readers would appreciate more attention to previous findings and a clearer outline of how this work differs. Two points stand out, outlined below. I believe the authors can address these potential complaints in the introduction and discussion, and perhaps provide some clarification in the presentation of the results.In the introduction, the authors state that "... to date, no study has been able to provide single-trial evidence of multiple EEG components involved in decision-making..." (line 64). Many readers would disagree with this. For example, Philiastides, Ratcliff, & Sadja (2006) use a single-trial analysis to unravel early and late EEG components relating to decision difficulty and accuracy (across different perceptual decisions), which could be related to the components in the current work. Other, network-based single-trial EEG analyses (e.g., Si et al., 2020, NeuroImage, Sturn et al., 2016 J Neurosci Methods) could also be related to the current component approach. Yet other approaches have used inverse encoding models to examine EEG components related to separable decision processes within trials (e.g., Salvador et al., 2022, Nat Comms). The results of the current work are consistent with this previous work - the two components from Philiastides et al., 2006 can be mapped onto the components in the current work, and Salvador et al., 2022 also uncover stimulus- and decision-deliberation related components.

We completely agree with the reviewer that the link to previous work was insufficient. We now include all references that the reviewer points out both in the introduction (see response R3.2) and in the discussion (see response R3.4). We wish to thank the reviewer for bringing these papers to our attention as they are important for the manuscript.

The authors relate their components to ERPs. This prompts the question of whether we would get the same results with ERP analyses (and, on the whole, the results of the current work are consistent with conclusions based on ERP analyses, with the exception of the missing motor component). It's nice that this analysis is single-trial, but many of the follow-up analyses are based on grouping by condition anyway. Even the single-trial analysis presented in Figure 4 could be obtained by median splits (given the hypotheses propose opposite directions of effects, except for the linear model).

We do not agree with the reviewer in the sense that classical ERP analyses would require much more data-points. The performance of the method is here to use the information shared across all contrast levels to be able to model the processing time of a single contrast level (6 trials per participant). Furthermore, as stated in the response to R1.4 and R1.5, the aim of the paper is to have the time of information processing components which cannot be achieved with classical ERPs without strong, and likely false, assumptions.

Medium Comments:(1) The presentation of Piéron's law for the behavioural analysis is confusing. First, both laws should be clearly defined for readers who may be unfamiliar with this work. I found the proposal that Piéron's law predicts decreasing RT for increasing pedestal contrast in a contrast discrimination paradigm task surprising, especially given the last author's previous work. For example, Donkin and van Maanen (2014) write "However, the commonality ofPiéron's Law across so many paradigms has lead researchers (e.g., Stafford & Gurney, 2004; Van Maanen et al., 2012) to propose that Piéron's Law is unrelated to stimulus scaling, but is a result of the architecture of the response selection (or decision making) process." The pedestal contrast is unrelated to the difficulty of the contrast discrimination task (except for the consideration of Fechner's law). Instead, Piéron's law would apply to the subjective difference in contrast in this task, as opposed to the pedestal contrast. The EEG results are consistent with these intuitions about Piéron's law (or more generally, that contrast is accumulated over time, so a later EEG component for lower pedestal contrast makes sense): pedestal contrast should lead to faster detection, but not necessarily faster discrimination. Perhaps, given the complexity of the manuscript as a whole, the predictions for the behavioural results could be simplified?

We agree that the initial version was confusing. We now clarified the presentation of Piéron's law at the end of the introduction (see also response to R2).

Once Fechner's law is applied, decision difficulty increases with increasing contrast, so Piéron's law on the decision-relevant intensity (perceived difference in contrast) would also predict increasing RT with increasing pedestal contrast. It is unlikely that the data are of sufficient resolution to distinguish a log function from a power of a log function, but perhaps the claim on line 189 could be weakened (the EEG results demonstrate Piéron's law for detection, but do not provide evidence against Piéron's law in discrimination decisions).

This is an excellent observation, thank you for bringing it to our attention. Indeed, the data support the notion that Pieron’s law is related to detection, but do not rule out that it is also related to decision or discrimination. In earlier work, we (Donkin & Van Maanen, 2014) addressed this question as well, and reached a similar conclusion. After fitting evidence accumulation models to data, we found no linear relationship between drift rates and stimulus difficulty, as would have been the case if Pieron's law could be fully explained by the decision process (as -indirectly- argued by Stafford & Gurney, 2004; Van Maanen et al., 2012). The fact that we observed evidence for a non-linear relationship between drift rates and stimulus difficulty led us to the same conclusion, that Pieron’s law could be reflected in both discrimination and decision processes. We added the following comment to the discussion about the functional locus of Pieron's law to clarify this point:

“Note that this observation is not necessarily incongruent with theoretical work that argued that Piéron’s law could also be a result of a response selection mechanism (Stafford and Gurney, 2004; Van Maanen et al., 2012; Palmer et al., 2005). It could be that differences in stimulus intensity between the two options also contribute to a Piéron like relationship in the later intervals, that is convoluted with Fechner’s law (see Donkin and Van Maanen, 2014, for a similar argument). Unfortunately, our data do not allow us to discriminate between a pure logarithmic growth function and one that is mediated by a decreasing power function”.

(2) Appendix 1 shows that the event detection of the HMP method will also pick up on ramping activity. The description of the problem in the introduction is that event-like activity could look like ramping when averaged across trials. To address this problem, the authors should simulate events (with some reasonable dispersion in timing such that they look like ramping when averaged) and show that the HMP method would not pull out something that looked like ramping. In other words, the evidence for ramping in this work is not affected by the previously identified confounds.

We agree that this demonstration was necessary and thus added the suggested simulation to Appendix 1. As can be seen in the Figure 1 of the appendix, when we simulate a half-sine the average ERP based on the timing of the event looks like a half-sine.

(3) Some readers may be interested in a fuller discussion of the failure of the Fechner diffusion model in the speed condition.

We are unsure which failure the reviewer refers to but assumed it was in relation to the behavioral results and thus added:

It is unlikely that neither Piéron nor Fechner law impact the RT in the speed condition. Instead this result is likely due to the composite nature of the RT where both laws co-exist in the RT but cancel each other out due to their opposite prediction.

Minor Comments:(1) "By-trial" is used throughout. Normally, it is "trial-by-trial" or "single-trial" or "trial-wise".

We replaced all occurrences of “by-trial” with the three terms suggested were appropriate.

(2) Line 22: "The sum of the times required for the completion of each of these precessing steps is the reaction time (RT)." The total time required. Processing.

Corrected for both.

(3) Line 26/27: "Despite being an almost two century old problem (von Helmholtz, 2021)." Perhaps the citation with the original year would make this point clearer.

We agree and replaced the citation.

(4) Line 73: "accounted by estimating". Accounted for by estimating.

Corrected.

(5) Line 77 "provides an estimation on the." Of the.

Corrected.

(6) Line 86: "The task of the participants was to answer which of two sinusoidal gratings." The picture looks like Gabor's? Is there a 2d Gaussian filter on top of the grating? Clarify in the methods, too.

We incorrectly described the stimuli as those were indeed just Gabor’s. This is now corrected both in the main text and the method section.

(7) Figure 1 legend: "The Fechner diffusion law" Fechner's law or your Fechner diffusion model?

Law was incorrect so we changed to model as suggested.

(8) Line 115: "further allows to connects the..." Allows connecting the.

Corrected.

(9) Line 123: "lower than 100 ms or higher than..." Faster/slower.

Corrected.

(10) Line 131: "To test what law." Which law.?

Corrected to model.

(11) Figure 2 legend: "Left: Mean RT (dot) and average fit (line) over trials and participants for each contrast level used." The fit is over trials and participants? Each dot is? Average trials for each contrast level in each participant?

This sentence was corrected to “Mean RT (dot) for each contrast level and averaged predictions of the individual fits (line) with Accuracy (Top) and Speed (Bottom) instructions.”.

(12) Line 231: "A comprehensive analysis of contrast effect on". The effect of contrast on.

This title was changed to “functional interpretation of the events”.

(13) Line 23: "the three HMP event with". Three HMP events.

The sentence no longer exists in the revised manuscript.

(14) Line 270: "Secondly, we computed the Pearson correlation coefficient between the contrast averaged proportion of correct." Pearson is for continuous variables. Proportion correct is not continuous. Use Spearman, Kendall, or compute d'.

The reviewer rightly pointed out our error, we corrected this by computing Spearman correlation.

(15) Line 377: "trial 𝑛 + 1 was randomly sampled from a uniform distribution between 0.5 and 1.25 seconds." It's just confusing why post-response activity in Figure 5 does look so consistent. Throughout methods: "model was fitted" should be "was fit", and line 448, "were split".

We do not have a specific hypothesis of why the post-response activity in the previous Figure 5 was so consistent. Maybe the Gaussian window (same as in other manuscripts with a similar figure, e.g. O’Connell et al. 2012) generated this consistency. We also corrected the errors mentioned in the methods.

(16) The linear mixed models paragraph is a bit confusing. Can it clearly state which data/ table is being referred to and then explain the model? "The general linear mixed model on proportion of correct responses was performed using a logit link. The linear mixed models were performed on the raw milliseconds scale for the interval durations and on the standardized values for the electrode match." We go directly from proportion correct to raw milliseconds...

The confusion was indeed due to the initial inclusion of a general linear mixed model on proportion correct which was removed as it was not very informative. The new revision should be clearer on the linear mixed models (see first sentence of subsection ‘linear mixed models' in the method section).

(17) A fuller description of the HMP model would be appreciated.

We agree that this was necessary and added the description of the HMP model in the corresponding method section “Hidden multivariate pattern” in addition to a more comprehensive presentation of HMP in the first paragraph of the Result and Discussion sections.

(18) Line 458: "Fechner's law (Fechner, 1860) states that the perceived difference (𝑝) between the two patches follows the logarithm of the difference in physical intensity between..." ratio of physical intensity.

Corrected.

(19) P is defined in equations 2 and 4. I would include the beta in equation 4, like in equation 2, then remove the beta from equations 3 and 5 (makes it more readable). I would also just include the delta in equation 2, state that in this case, c1 = c+delta/2 or whatever.

This indeed makes the equation more readable so we applied the suggestions for equations 2, 3, 4 and 5. The delta was not added in equation 2 but instead in the text that follows:

“Where 𝐶1 = 𝐶0 + 𝛿, again with a modality and individual specific adjustment slope (𝛽).”

(20) The appendix suggests comparing the amplitudes with those in Figure 3, but the colour bar legend is missing, so the reader can only assume the same scale is used?

We added the color bar as it was indeed missing. Note though that the previous version displayed the estimation for the simulated data while this plot in the revised manuscript shows the solution on real data obtained after downsampling the data (and therefore look for a larger pattern as in the main text). We believe that this representation is more useful given that the solution for the downsampled data is no longer the same as the one in the main text (due to the difference in pattern width).